# Identifying species likely threatened by international trade on the IUCN Red List can inform CITES trade measures

Daniel W. S. Challender [1] ✉, Patricia J. Cremona [2], Kelly Malsch[3], Janine E. Robinson[4,5], Alyson T. Pavitt[3], Janet Scott[2], Rachel Hoffmann[6,7], Ackbar Joolia[2], Thomasina E. E. Oldfield[8,9], Richard K. B. Jenkins[2], Dalia A. Conde [10,11], Craig Hilton-Taylor [2] & Michael Hoffmann [12]

Overexploitation is a major threat to biodiversity and international trade in many species is regulated through the Convention on International Trade in Endangered Species of Wild Fauna and Flora (CITES). However, there is no established method to systematically determine which species are most at risk from international trade to inform potential trade measures under CITES. Here, we develop a mechanism using the International Union for Conservation of Nature's Red List of Threatened Species to identify species that are likely to be threatened by international trade. Of 2,211 such species, CITES includes 59% (1,307 species), leaving two-fifths overlooked and in potential need of international trade regulation. Our results can inform deliberations on potential proposals to revise trade measures for species at CITES Conference of the Parties meetings. We also show that, for taxa with biological resource use documented as a threat, the number of species threatened by local and national use is four times greater than species likely threatened by international trade. To effectively address the overexploitation of species, interventions focused on achieving sustainability in international trade need to be complemented by commensurate measures to ensure that local and national use and trade of wildlife is well-regulated and sustainable.

Preventing the overexploitation of species (harvesting at a rate that exceeds the ability of populations to recover) requires knowledge of the species, the associated harvest and trade levels and the impact on populations and, where necessary, implementation of proportionate interventions at local, national and, if relevant, global scales[1,2]. These may variously include sustainable management programmes, supply-side measures (for example, commercial captive breeding), increased law enforcement and supportive national and international

[1]Interdisciplinary Centre for Conservation Science (ICCS), Department of Biology and Oxford Martin School, University of Oxford, Oxford, UK. [2]IUCN Science & Data Centre: Biodiversity Assessment & Knowledge Team, The David Attenborough Building, Cambridge, UK. [3]UN Environment Programme World Conservation Monitoring Centre (UNEP-WCMC), Cambridge, UK. [4]Durrell Institute of Conservation and Ecology (DICE), School of Anthropology and Conservation, University of Kent, Canterbury, UK. [5]Joint Nature Conservation Committee (JNCC), Peterborough, UK. [6]Sustainable Use and Livelihoods Specialist Group, Species Survival Commission/Commission on Environmental, Economic and Social Policy, International Union for Conservation of Nature (IUCN), Gland, Switzerland. [7]International Institute for Environment and Development (IIED), London, UK. [8]TRAFFIC, The David Attenborough Building, Cambridge, UK. [9]Independent Consultant, Cambridge, UK. [10]Species360 Conservation Science Alliance, Bloomington, MN, USA. [11]Interdisciplinary Centre on Population Dynamics, Department of Biology, University of Southern Denmark, Odense, Denmark. [12]Conservation and Policy, Zoological Society of London, London, UK. ✉e-mail: dan.challender@biology.ox.ac.uk

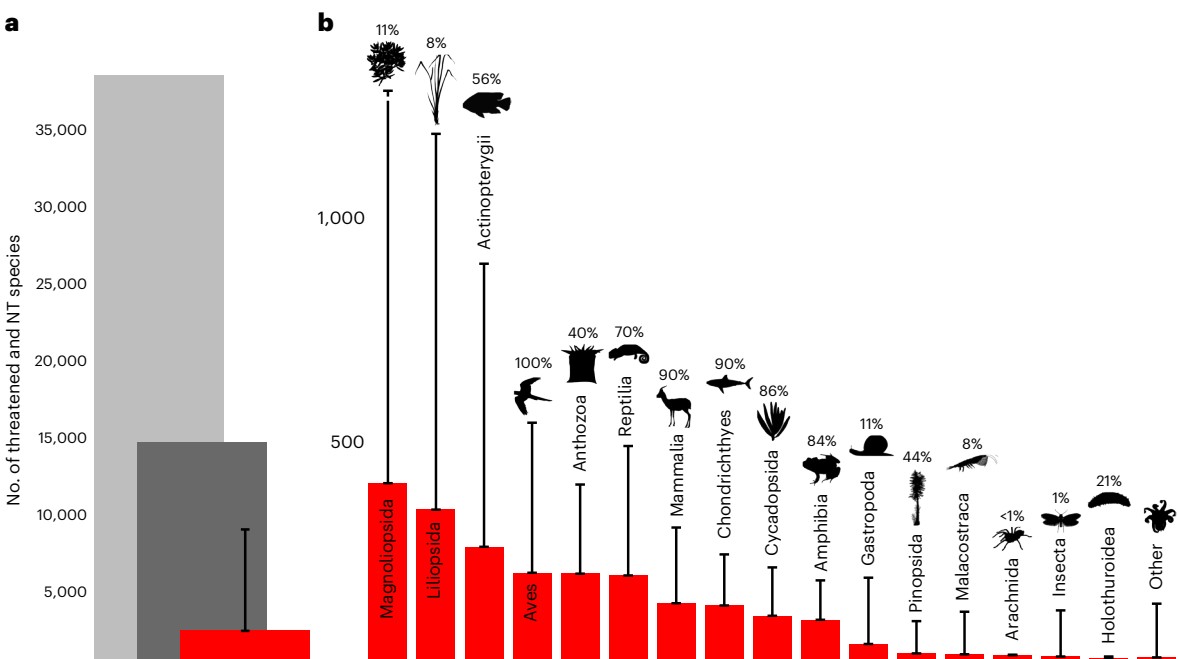

**Fig. 1 | Threat status. a**, Number of threatened and Near Threatened (NT) species on the Red List (38,245; light grey), species with any coded BRU threats included in our dataset (14,741; dark grey) and species likely to be threatened by international trade (2,211; red). **b**, Number of threatened and Near Threatened species on the Red List likely to be threatened by international trade, by class. 'Other' includes classes with fewer than ten species likely to be threatened by international trade. Error bars in **a** and **b** represent lower and upper bounds accounting for the uncertainty of species categorized under insufficient information (Supplementary Methods 2.5). In **b**, the upper bound for Magnoliopsida is 3,369 species. Percentages indicate proportion of species in each class assessed on the Red List. Credit for *Gazella gazella* image: Rebecca Groom, under a Creative Commons license CC BY 3.0 (without changes).

policies among others[3]. The Convention on International Trade in Endangered Species of Wild Fauna and Flora (CITES), which entered into force in 1975, seeks to ensure that international trade in wildlife is ecologically sustainable, as well as legal and traceable, and regulates trade in ~39,000 species, most of which (85%) are plants[4]. Although focused on regulating legal international trade, the treaty has had to contend with illegal trade due to the well-publicized detrimental impact of such trade on species[5]; between 2010 and 2018 at least US$2.3 billion was spent on combatting wildlife trafficking globally[6].

Of species currently included in CITES, most were added to the Convention after its inception at triennial Conference of the Parties (CoP) meetings. Decisions are made at these meetings on, inter alia, the establishment, removal and amendment of trade controls for hundreds, sometimes thousands, of species. These measures correspond with the listing of species in one of three appendices and are implemented through national legislation and a system of permits and certificates. Nearly 1,100 species[4] are included in Appendix I of CITES, having been deemed threatened with extinction and which are (or may be) affected by trade and in which commercial, international trade is prohibited. Most species (~37,000; ref. 4) are included in Appendix II, trade in which is closely regulated. Appendix III includes species in which trade is regulated by one country but it requires international cooperation in doing so. International trade in CITES-listed species is subject to a declaration by exporting countries (and importing countries for Appendix I species) that it is not detrimental to wild populations (the non-detriment finding (NDF)) and is legal (the legal acquisition finding). Listing criteria have been adopted against which proposed amendments to the appendices are evaluated on the basis of an assessment of biological and trade data[7]. They allow for the listing of entire groups (higher-taxon listings; for example, the ~28,000 orchids (Orchidaceae spp.)) and species that resemble other taxa in trade (or look-alike species). Proposed amendments to the appendices are adopted by consensus or subject to a two-thirds majority vote by parties.

The approach taken to table proposals at CoPs is far from systematic. Proposals must be submitted to the CITES Secretariat at least 150 days before CoPs, can only be submitted by parties and are typically submitted by range countries for particular species. The adoption of proposals depends on the weight of evidence in the proposal and whether there is strong support or opposition from parties. The latter may depend on the profile of the species (for example, iconic species tend to receive support for trade restrictions) and, relatedly, whether species are championed by parties, non-governmental organizations (NGOs) and lobby groups[8]. In reality, some countries, which may not have the necessary resources or relevant scientific expertise, collaborate with NGOs to develop proposals, while NGOs and other groups may also draft proposals independently and seek out parties receptive to their submission[8]. The Convention's depository government (Switzerland) usually submits a few proposals (following recommendations from the CITES scientific committees (Animals and Plants Committees)) as does the meeting host country. This results in a geographically and taxonomically diverse range of proposals which ostensibly represent national, regional and other stakeholder (for example, NGO) priorities. However, this approach may not reliably identify those species that are most threatened by international trade or in greatest need of better trade regulation and it is likely that many at-risk species are being overlooked.

Using The IUCN Red List of Threatened Species[9] (hereafter, Red List), widely acknowledged as the most authoritative source on extinction risk and threats to species globally, we provide a systematic assessment of the likelihood of threat posed by international trade across all taxonomic groups (Methods, Supplementary Context and Supplementary Methods 2.1–2.8). Starting with >38,000 globally threatened and Near Threatened species on the Red List (version 2020-1), we used selection criteria to identify species potentially threatened by international trade. The selection criteria comprised threatened (species that are categorized as Critically Endangered, Endangered or Vulnerable)

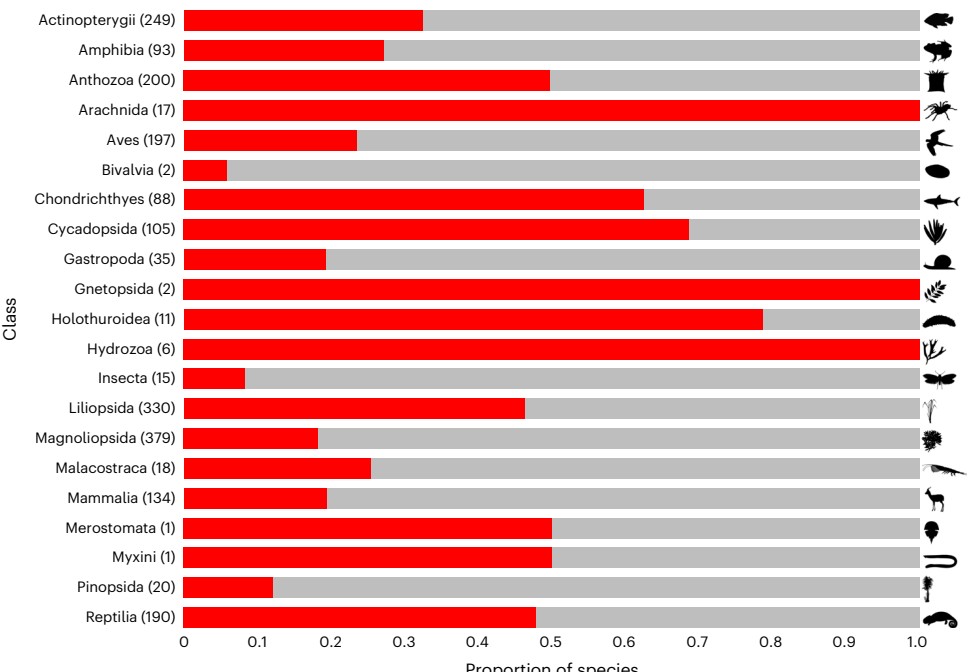

**Fig. 2 | Threat scale.** Threatened and Near Threatened species categorized as likely to be threatened by international trade (red) as a proportion of species in these threat categories included in our dataset with BRU threats coded (grey), by class. Numbers in parentheses are species likely to be threatened by international trade. Excludes 17 species likely to be threatened by international trade that do not have BRU threats coded. Credit for *Gazella gazella* image: Rebecca Groom, under a Creative Commons license CC BY 3.0 (without changes).

and Near Threatened threat categories, relevant threat codes, the presence of particular terms within assessments (for example, commercial use) and information on the scale of end-uses for species (for example, subsistence or international) (Methods). We subsequently categorized the resulting 21,745 species as 'likely' or 'unlikely' to be threatened by international trade or as having 'insufficient information' to determine the likelihood of this threat (Methods; Extended Data Fig. 1). We then identified which of these species are, and are not, included in the CITES appendices and evaluated the results in the context of threats to species from biological resource use (BRU), including comparing species likely to be threatened by international trade with those considered threatened by use and/or trade at the local and/or domestic level on the Red List.

## Results and discussion

Of 38,245 globally threatened and Near Threatened species, 5.8% (2,211 species) are likely to be threatened by international trade (Fig. 1a, Extended Data Table 1, Supplementary Table 1 and Supplementary Data 1). Incorporating uncertainty from those species categorized as having insufficient information, the proportion is between 5.8% and 23% (8,796 species: midpoint 15%, 5,737 species) (Fig. 1a, Supplementary Methods 2.5 and Supplementary Results 4.1–4.2). Of the 2,211 species, nearly half (47%, $n = 1,041$) face an extremely high (Critically Endangered) or very high (Endangered) risk of extinction (Extended Data Fig. 2) with international trade as a contributing factor. Recognizing variation in the proportion of species in each class that have been assessed on the Red List, our results indicate that one-third of all species likely to be threatened by international trade are plants in the classes Magnoliopsida ($n = 402$, mainly cacti (Cactaceae spp.), dipterocarps (Dipterocarpaceae spp.) and legumes (Fabaceae spp.)) and Liliopsida ($n = 343$, predominantly orchids) (Fig. 1b). Other notable classes include ray-finned fishes (Actinopterygii; $n = 260$), various birds (Aves; $n = 202$, about one-third of which are parrots (Psittacidae spp.)), anthozoans (Anthozoa; $n = 200$, mainly stony corals (for example,

Acroporidae spp.)) and reptiles (Reptilia; $n = 196$) among other diverse groups (Fig. 1b).

More than two-thirds (68%, 14,741 of 21,745 species) of the threatened and Near Threatened species that met our selection criteria have one or more forms of BRU—whether intentional or unintentional—documented as a recognized threat. Of these taxa, the proportion of species in individual classes that are likely to be threatened by international trade ranges from <10% in some groups (for example, bivalves (Bivalvia)) to half or more in others: fire corals (Hydrozoa: Milleporidae; 6 of 6 assessed species), sea cucumbers (Holothuroidea; 11 of 14 species), arachnids (Arachnida; 18 of 22 species), cycads (Cycadopsida; 105 of 167 species) and anthozoans (200 of 403 species) (Fig. 2). These proportions increase for most groups when species which only have unintentional uses documented are excluded (Extended Data Fig. 3 and Supplementary Results 4.2).

More than half (59%, $n = 1,307$) of the species determined to be likely threatened by international trade are listed in one or more of the CITES appendices (Fig. 3a and Extended Data Table 2). This suggests that the Convention performs moderately well at capturing species that are or may be affected by international trade or are otherwise in need of trade regulation, especially considering the lack of a readily accessible evidence base to date on species threatened by overexploitation for international trade. These 1,307 species include taxa facing an extremely high (Critically Endangered; $n = 295$) or very high (Endangered; $n = 376$) risk of extinction (Extended Data Table 3), which are being negatively impacted by international trade and/or trafficking, therefore warranting concerted conservation attention at local to global scales. Examples are pangolins (Manidae spp.)[10] and the European eel (*Anguilla anguilla*)[11], which are trafficked for human consumption, and various orchids (for example, *Paphiopedilum* spp.)[12] and cycads (for example, *Encephalartos* spp.)[13], which are used for horticulture, food and medicine. All hydrozoans, anthozoans and most arachnids that are likely to be threatened by international trade are included in CITES (Extended Data Table 2).

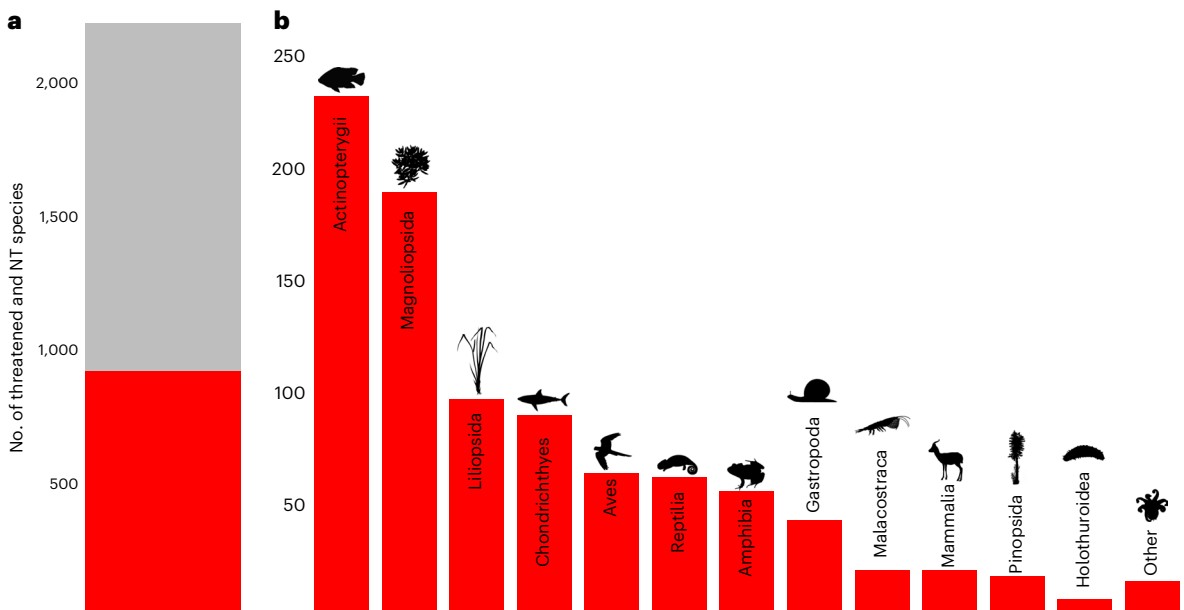

**Fig. 3 | CITES status. a**, Number of threatened and Near Threatened (NT) species on the Red List likely to be threatened by international trade included (grey; 1,307 species) and not included (red; 904) in CITES. **b**, Number of threatened and Near Threatened species on the Red List that are likely to be threatened by international trade but are not included in CITES (904), by class. 'Other' includes classes with fewer than seven species. Credit for *Gazella gazella* image: Rebecca Groom, under a Creative Commons license CC BY 3.0 (without changes).

Although our analyses do not capture the length of time that species have been included in CITES, that 1,307 species are likely to be threatened by international trade despite being included in the Convention suggests that greater scrutiny of the implementation and effectiveness of CITES is needed. These taxa include species listed in Appendices I, II and III, 81% of which (*n* = 1,063) are globally threatened with extinction (Extended Data Tables 2 and 3). Our results can inform decision-making in CITES; in particular, close attention should be given to NDFs for these species by parties. They also highlight Appendix II-listed species which may warrant inclusion in the review of significant trade process (subject to meeting the criteria)[14] and Appendix I-listed species that may benefit from ad hoc reviews and associated recommendations[15,16]. They further highlight species that could potentially benefit from additional trade controls (for example, export quotas or transfer from Appendix II to I) and/or other interventions. Whether and which measures may be needed for particular species will depend on the scope (that is, the proportion of the population affected; for example, a single subpopulation or most of the global population) and severity of the threat (for example, the population declines caused by the threat) (Supplementary Discussion 5.1) and the probable effectiveness of any measures considering the social-ecological systems (SESs) in which the harvest, use and trade of species occur[17] (see section on Solutions in global data).

This leaves 41% (904 species) that are likely to be threatened by international trade and not currently listed in CITES (Fig. 3b, Extended Data Tables 2 and 4 and Supplementary Table 1). Of these species, 41% (*n* = 370) are Critically Endangered or Endangered. Overall, >25% of these 904 species not included in CITES are ray-finned fishes (*n* = 231, notably cichlids (Cichlidae spp.) and carps (Cyprinidae spp.)) and >20% are plants in the class Magnoliopsida (*n* = 188), many of which are dipterocarps. Other major groups include the class Liliopsida (*n* = 95, for example, palms (Arecaceae spp.)), cartilaginous fishes (*n* = 89, for example, requiem sharks (Carcharhinidae spp.)), birds (*n* = 63), reptiles (*n* = 61) and amphibians (Amphibia; *n* = 55) among others. All 904 species should be of interest to the CITES parties because they are likely at some risk from international trade, depending on the scope and severity of threat from offtake for international trade (see criteria in

Methods and Supplementary Discussion 5.1) and therefore may benefit from commensurate regulatory measures. The CITES CoP19 meeting in November 2022 adopted proposals to include numerous shark species in the appendices, including many highlighted in our analyses. Critically Endangered and Endangered species are obvious priorities for further evaluation of the impact of international trade because they face a higher extinction risk. These results can inform deliberations on potential proposals to revise trade measures for species ahead of CITES CoPs and can highlight overlooked taxonomic groups that may warrant greater attention under the Convention.

## Solutions in global data

CITES is a scientific Convention but the approach taken to propose trade controls for species—the principal tenet of the Convention—is unsystematic and a more systematic approach would help to ensure that high-risk species are afforded appropriate international trade measures where they would benefit in conservation terms[17]. The Red List and the methods presented here for rapid, systematic risk assessment offer a first step in identifying a subset of priority species that may warrant further consideration against the CITES listing criteria (Supplementary Discussion 5.1–5.2). The Red List contains assessments of extinction risk for >142,000 species and has a goal of assessing an additional 129,000 species by 2030 (ref. 18). New assessments are added to the Red List through multiple updates a year and are complemented by reassessments with a target of reassessing each species every 4–10 years. While the Red List categories and criteria differ from the CITES listing criteria (Supplementary Methods 2.6), future iterations of our results, which focus on new and updated assessments, could be shared with the parties and other stakeholders to inform potential proposals to amend the appendices. We have demonstrated that our mechanism for categorizing species can be fully automated and can produce results comparable to a person manually assigning species to a category (Methods; Supplementary Methods 2.4 and 2.7, Supplementary Tables 3–6 and Supplementary Results 4.3), meaning the results could be produced rapidly and shared with the parties when needed (for example, at CoP or Animals and Plants Committee meetings; Supplementary Discussion 5.1). They could inform potential proposals

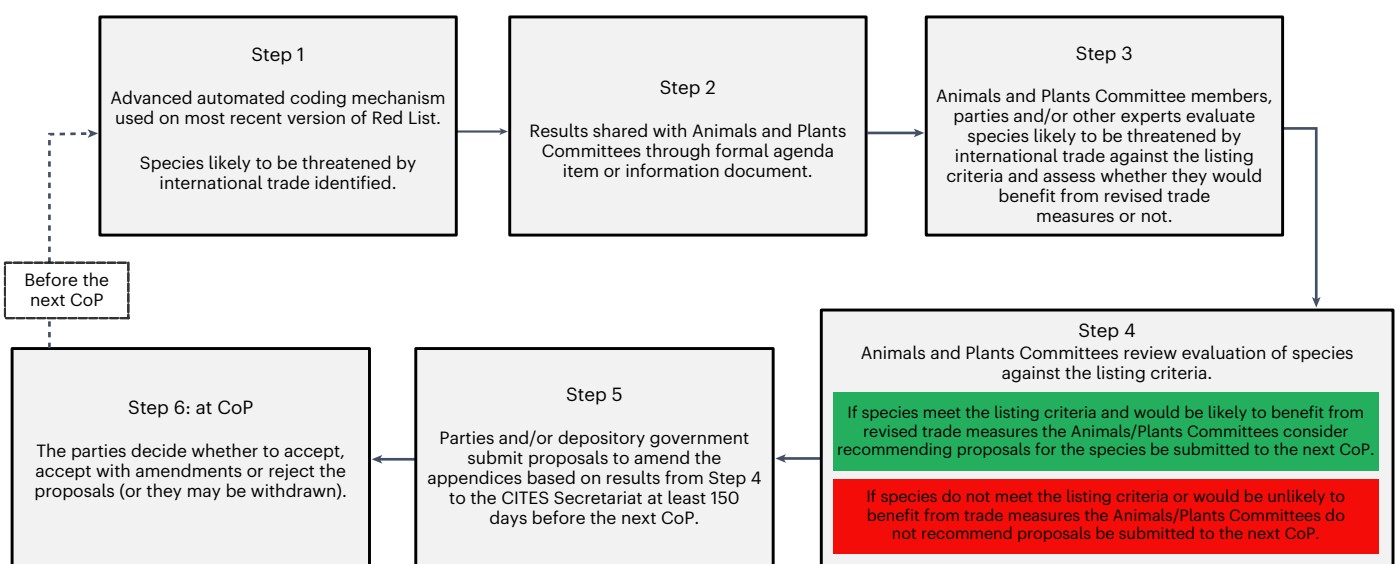

**Fig. 4 | Proposed process for integration of results into decision-making by the CITES scientific committees.** See also Supplementary Discussion 5.1 and 5.2.

to amend the appendices in two ways. First, by providing a starting point for parties to proactively develop proposals for species that are likely to be negatively impacted by international trade. Second, they could inform the Convention's scientific committees about species that merit further examination between CoPs. At CoP19, the parties adopted Decision 19.186, which directs the CITES scientific committees to consider mechanisms through which to provide parties with information on species that may warrant international trade regulation. Recognizing the need for discussion with, and agreement from, the scientific committees (Supplementary Discussion 5.2), species could then be subsequently evaluated against the listing criteria, including drawing on additional information sources beyond the Red List, to determine which, if any, criteria they meet (Fig. 4). Data on species already included in CITES and categorized as likely to be threatened by international trade in future analyses could also be shared with the parties and scientific committees at this time for their consideration of any further trade and/or conservation measures that may be needed.

Where species are considered to meet the listing criteria, proposing parties and/or the Animals/Plants Committees should explicitly evaluate whether the proposed measures would realistically be expected to contribute to the conservation of the species, or not, and any associated risks[17]. This is critically important because, while it is difficult to predict the effectiveness of CITES trade measures, they may sometimes do more harm than good for species (for example, by removing conservation incentives[19] or lead to accelerated wild harvest of species[20,21]). Parties should consider assessing how and why particular outcomes may be expected on the basis of an understanding of the relevant SESs, including how harvest incentives may change, how actors along supply chains may respond and any likely adverse impacts[17]. Uncertainty could be further reduced by parties identifying additional measures that would be needed to mitigate any identified risks and support the implementation of trade measures. This could include, for example, greater resources for law enforcement agencies to ensure adequate probabilities of apprehension for would-be offenders, the establishment of partnerships with local communities to sustainably manage species and/or programmes to change consumer behaviour[22]. Where species would be likely to benefit from trade measures, parties could submit proposals to the next CoP and the scientific committees could also recommend that proposals be submitted to these meetings (Fig. 4). This process would complement the submission of proposals based on other priorities (for example, national and/or NGO priorities).

The adoption of proposals to amend the appendices emerging from the mechanism presented here would establish, or increase, international trade controls for species. Acknowledging the difficulty of predicting the impact of these measures, where trade controls are successfully combined with other supportive interventions, they can contribute to positive conservation outcomes. For example, the sustainable use of rhinoceroses in parts of Africa[23] and numerous crocodilians in different parts of the world[24], as well as conservation of the greater one-horned rhinoceros (*Rhinoceros unicornis*) in Asia, populations of which are increasing[25]. Where trade controls appear ineffective, CITES has additional processes designed to prevent detrimental international trade, ensure compliance among parties and catalyse implementation of supportive interventions[26]. These include requirements for NDFs, the review of significant trade process (a species-specific non-compliance response mechanism) for Appendix II-listed species[14], review mechanisms for Appendix I-listed species[16] and bespoke measures for particular species and groups, including those agreed through resolutions and decisions. Species-focused resolutions, among other things, typically encourage parties and other stakeholders to implement interventions that address the drivers of unsustainable harvest and trade more directly (for example, by engaging local communities in the management of species and/or changing consumer behaviour)[27–29]. Where compliance issues remain, stricter mechanisms exist, including the use of trade suspensions, the use of Article XIII measures (a process through which recommendations are made to ensure effective implementation of the treaty by particular parties)[26] and the use of political and diplomatic means to ensure parties are complying and fully implementing the provisions of the Convention[30].

Importantly, systematic threat assessments need not be restricted to identifying species that may warrant greater trade regulation. They could equally inform the relaxation of trade controls for species that have improved in status and can potentially be traded on a sustainable basis (Supplementary Discussion 5.1). For example, the transfer of the Cape mountain zebra (*Equus zebra zebra*) from Appendix I to II in 2016.

We caution that species herein determined to be likely threatened by international trade may not necessarily meet the CITES listing criteria[7]; they would need to be evaluated on a case-by-case basis alongside relevant information from other data sources (Fig. 4 and Supplementary Discussion 5.1). The species in this category were so included because they met the relevant criteria developed (Methods) but their inclusion in this category does not imply that international

trade constitutes a major threat or that the threat applies throughout the species' geographic range (Supplementary Discussion 5.1). These results also rely only on the information contained in the Red List, which has limitations and in certain cases may need updating, and this has implications for how the Red List records data (Supplementary Discussion 5.3). Despite these limitations, the Red List is currently the most comprehensive source of information available on the degree of threat to species from international trade.

## Threats in context

Previous studies[1,31–33] have consistently shown that unsustainable hunting and collecting are major threats to biodiversity. Ensuring that species threatened by international trade are identified and international trade controls established, where such species would be likely to benefit in conservation terms, is a crucial step to safeguarding species from overexploitation. A recent study[34] presented an analysis of Red List data to understand the extent to which use of wild species is, or is not, having a detrimental impact on species extinction risk but the study did not consider the geographic scale at which this use takes place. Our results specifically examine which species are likely to be threatened by international trade and suggest that many more species are threatened by use and trade at a local and/or national (domestic) level (Methods). Of 14,741 globally threatened or Near Threatened species that have BRU as a threat on the Red List, 15% (2,194 species) are likely to be threatened by international trade (Supplementary Table 2). Incorporating uncertainty regarding species categorized as having insufficient information, the proportion is between 15% and 44% (6,486 species: midpoint 21%, 3,096 species) (Supplementary Methods 2.5 and Supplementary Results 4.1). Taking the midpoint suggests that around one in five species (3,096 of 14,741, 21%) that have BRU as a threat is likely to be threatened by international trade and the remaining 79% are threatened by use and trade that is local and/or domestic in scale. These results suggest that the response of governments and the international donor community to combatting unsustainable and illegal international trade needs to be complemented by an even greater commitment to mitigating threats from unsustainable use and trade at local and domestic levels. This will necessitate context-specific interventions cocreated between local and national stakeholders and may varyingly include sustainable-use programmes, further regulations on the harvest and domestic trade of species, partnerships with rural communities and the private sector, commercial captive breeding, effective site-based protection and good governance along supply chains[3,5]. Robust management plans will be essential having been shown to be key to achieving conservation goals[35] but which are lacking for many species threatened by use and/or trade[34]. Finally, future iterations of our analyses could explicitly indicate CITES-listed species that are threatened by local and/or domestic use and/or trade, rather than exploitation for international trade, and be shared with the CITES parties to inform appropriate actions, including scrutiny of NDFs (Supplementary Data 2).

More broadly, as the world's governments convene to set ambitious nature protection targets for the Post-2020 Global Biodiversity Framework[36], the mechanism presented here could be used for tracking progress towards international goals to eliminate the negative impacts of unsustainable harvest for international trade on biodiversity.

Cross-referencing data from the Red List with CITES listing information is a valuable method for estimating the prevalence of threat to species from international trade and generating insights into CITES trade measures, including identification of potential gaps. The mechanism presented here can ensure that the international community has a more robust evidence base to inform decision-making on establishment or adjustment of international trade controls in the future—supporting the CITES Strategic Vision[37] (Supplementary Discussion 5.2)—while simultaneously contributing to the assessment of global efforts to conserve biodiversity.

## Methods

### Species selection

We used data for 38,245 threatened and Near Threatened species from Red List version 2020-1 and coded species to assign them to a category pertaining to threat from international trade based on available information in Red List assessments (hereafter, assessments). Background on the Red List and limitations to using these data for this purpose are provided in Supplementary Methods 2.1–2.3.

To identify species that may be threatened by international trade we queried the Red List and constructed an MS Excel database of candidate species. A PostgreSQL database, which contains a copy of all data from current published assessments, was used for data extraction; we ran five SQL queries on this database using pgAdminIII (database querying software). We used the combined results to assign species to a category using automated and manual coding (see below).

**Query 1.** The first query extracted the threat category and all data from the rationale, threats and use and trade sections (text fields) of assessments, for species selected on the basis of the following criteria: (1) species categorized as Critically Endangered (CR), Endangered (EN), Vulnerable (VU), Near Threatened (NT), Low Risk/near threatened (LR/nt) or Low Risk/conservation dependent (LR/cd); and either (2) assessments contained one or more of 53 text strings (for example, commercial use, full list in Supplementary Methods 2.2) within the rationale, threats and/or use and trade sections; or (3) assessments included one or more of 11 threat codes relating to BRU (5.1.1, 5.1.4, 5.2.1, 5.2.4, 5.3.1, 5.3.2, 5.3.5, 5.4.1, 5.4.2, 5.4.4 and 5.4.6; Supplementary Methods 2.2). Species classified as LR/nt and LR/cd were treated as NT, as per Red List guidance. We excluded Least Concern (LC) species on the basis that they are likely to be at lower risk from overexploitation and less likely to meet the CITES listing criteria. We also excluded Data Deficient (DD) species. This resulted in a database of 21,714 species.

The 53 text strings were chosen as those most likely to return species that may be threatened by international trade. We searched assessments using these text strings because for species listed as Extinct (EX), Extinct in the Wild (EW), CR, EN, VU, NT, LC and DD, it is a requirement when completing assessments that supporting information is provided in the threats text field in the form of a narrative on threats. For the species used, it is recommended, although not mandatory, that supporting information be included in the use and trade text field in the form of a narrative on use and trade.

Regarding threat codes, it is a requirement when completing assessments for species listed as EX, EW, CR, EN, VU and NT (but not LC or DD) that major threats to the species be coded according to the IUCN standardized Threats Classification Scheme[38]. We selected species where the threats included one or more of the 11 aforementioned threat codes on the basis that these species may be threatened by international trade. We included threat codes where motivation is unknown because, while the coding suggests that it is not known if the species is the target (of harvest), assessors are known to use this code when use is intentional but the scale is not known[34]. We included threat code 5.4.4 ((BRU) → Fishing & harvesting aquatic resources → Unintentional effects: large scale) because such species could theoretically be threatened by international trade, despite harvest being unintentional.

**Query 2.** The second query enabled us to add information from the IUCN Use and Trade Classification scheme[39] to our database, specifically the end-uses for which species were coded in the end-use table in assessments. On completing assessments for species that are used, it is recommended, although not mandatory, that supporting information on trade and/or use be included by means of indicating whether use is one or more of 'subsistence', 'national' and/or 'international'. Assessors are also asked to indicate the purpose of use from a list of 18 different purposes (for example, food—human; full list in Supplementary Methods 2.2). We used these data rather than the scale of use

(for example, local livelihood–subsistence) because doing so enabled us to distinguish between uses (at subsistence, national and/or international levels) comprising a threat to species and those that are not when combined with other information and applying our criteria to species.

We cross-referenced the results of our first two queries to identify any species that had any international uses coded but were not captured by our first query. This resulted in the addition of one species, *Cynanchum itremense*, to our database and 21,715 candidate species. See section on Species verification for detail on the process meaning our final dataset had 21,745 candidate species.

**Query 3.** The third query enabled us to add information to our database on whether international trade is recorded as a significant driver of threat to species. For a subset of threat codes (5.1.1, 5.2.1, 5.3.1, 5.3.2, 5.4.1 and 5.4.2) relating to intentional use, assessors are asked to code whether international trade is a significant driver of that threat to species, or not, or whether it is unknown. This code was only recently added to the data system, is not consistently applied and has only been used in a subset of assessments and therefore it is not yet a reliable indicator of the number of species threatened by international trade on the Red List. However, as data from this field can indicate whether international trade is a significant driver of threat for a subset of species, we included these data to aid categorization of species.

**Query 4.** The fourth query extracted data on coded threats to all species on the Red List, including whether threats were current, past or future; temporal data were added to our database for corresponding species. This enabled evaluation of coded threats to species relating to BRU.

**Query 5.** The fifth query extracted data from the IUCN Use and Trade Classification scheme[39] for candidate species, specifically from the field 'no use/trade information for this species'. This field is intended to be used to indicate that it is known or highly likely that the species is used and/or traded but further information is not available (Supplementary Methods 2.3).

### Species categorization

We developed criteria to assign species to a category—likely or unlikely to be threatened by international trade or insufficient information, adapting an approach developed by IUCN in 2015 (ref. 40). We applied the criteria to the 21,715 species that were selected using the process outlined above and using a combination of automated and manual coding (coding by a person). Our criteria are:

Species likely to be threatened by international trade:

(1) Intentional use is coded as a threat and 'is international trade a significant driver of threat' is coded as yes; or
(2) There is evidence to suggest that use and/or trade is a (probable or certain) threat to one or more populations/subpopulations (from threat code or description in rationale, threats or use and trade sections) and that form of use and/or trade is to some extent international (from international use being coded as yes and/or a relevant international end-use is coded and/or from description in rationale, threats or use and trade sections).

Insufficient information to determine if species is threatened by international trade:

(1) There is evidence to suggest that use and/or trade takes place (from threat codes or description in rationale, threats or use and trade sections or 'no use/trade information for this species' is coded as yes) and is a (probable or certain) threat to one or more populations/subpopulations (from threat codes or description in rationale, threats or use and trade sections) but there is no evidence that it is international and also no evidence

that it is not international (from description in rationale, threats or use and trade sections and international/national/subsistence uses not coded); or
(2) There is evidence to suggest that use and/or trade takes place (from threat codes or description in rationale, threats or use and trade sections or end-uses or 'no use/trade information for this species' is coded as yes), there is no evidence that it is not international (from description in rationale, threats or use and trade sections or international use is coded as yes) and either (i) there is no evidence that it is a threat and also no evidence that it is not a threat (from description in rationale, threats or use and trade sections) or (ii) it is described to be a past, future, potential, possible (or similar) threat; or
(3) There is no evidence that use or trade takes place (from threat codes or description in rationale, threats or use and trade sections and no uses are coded and 'no use/trade information for this species' is blank) but it is described as a potential future (or similar) threat.

Species unlikely to be threatened by international trade:

(1) There is no evidence that use or trade takes place (from threat codes or description in rationale, threats or use and trade sections, no end-uses are coded and 'no use/trade information for this species' is blank) and it is not described as a potential future (or similar) threat; or
(2) There is evidence to suggest that use and/or trade takes place (from threat codes or description in rationale, threats or use and trade sections, end-uses and 'no use/trade information for this species' is coded as yes) but that it is subsistence and/or national level and not international (from description in rationale, threats or use and trade sections or subsistence and/or national use coded as yes and international as no); or
(3) There is evidence to suggest that use and/or trade takes place (from threat codes or description in rationale, threats or use and trade sections, end-uses or no use/trade information for this species is coded as yes) but that it is not a threat (from description in rationale, threats or use and trade sections).

We took an evidentiary but precautionary approach (that is, assumed greater rather than lesser risk to species) to reasonably deduce from available information in each assessment whether international trade constitutes a threat to species or not. We focused on determining categorically whether there was evidence that international trade was a threat to species, regardless of the level of threat (Supplementary Methods 2.4). If we were unable to deduce from available information in each assessment that a species was threatened in any way by international trade, even if it is a species known to be impacted by international trade from other information sources, then it was categorized as 'insufficient information' or 'unlikely' on the basis of the information available. We used data on 'international trade is a significant driver of threat' (Query 3) to categorize species but did not use other responses ('no' and 'unknown') because the aim was to determine whether international trade posed any level of threat to species rather than being a significant driver of threat necessarily.

**Automated coding.** We coded 9,320 species to assign them to one of the three aforementioned categories using automated coding where it was feasible to do so based on the 'use and trade' and 'is international trade a significant driver of threat' fields and the relevance of use-related threat codes using R v.4.0.3 (ref. 41) (Extended Data Fig. 4 and Supplementary Methods 2.4). Species that were coded 'yes' for whether international trade is a significant driver of threat were coded 'likely'. Where the use and trade text field of assessments contained phrases such as 'information regarding the trade and use of this species is not known' or similar, the species was coded 'insufficient

information'. Where the use and trade text field included phrases such as 'there is no known use and trade in this species', or similar, the species was coded 'unlikely'. Where it was evident that species of fauna and funga had been included in our database based only on the presence of flora-related text strings (for example, 'timber') in assessments, they were categorized as 'unlikely'. These automation processes were tested extensively during development and were subsequently spot checked by a person for accuracy.

**Manual coding.** We manually coded the remaining 12,395 species to assign them a category because the available information needed to be interpreted by a human coder. This is because there is no direct link on the Red List between end uses and threats or scale of use beyond information in the text fields. Manual coding entailed reading the information and data for each assessment—text fields, threat codes, scale-of-use codes, purpose-of-use codes, 'no use/trade information on this species' field and 'is international trade a significant driver of threat' field—and categorizing species aided by a decision tree (Extended Data Fig. 5). For instance, a species with a relevant threat code may be used at the subsistence, national and/or international level and interpretation of the text fields was necessary to determine whether trade at the international level, rather than the subsistence and/or national level, comprised any level of threat (Supplementary Methods 2.4). Before coding, all coders trained on six batches of 100 randomly chosen species from our dataset. Before coding the full dataset, we measured our interrater reliability to ensure coders were categorizing species in a standardized way using 100 randomly selected amphibian species. We used Fleiss' Kappa in SPSS v.28 to test if agreement between all four coders was higher than would have been expected by chance. Parameter $\kappa = 0.85$ (95% CI, 0.79–0.91), $P < 0.0005$, indicating almost perfect agreement[42]. Remaining uncertainties were clarified among coders before coding the full dataset.

If a species could have been placed in one of two categories, we chose the most precautionary option; that is, assumed greater rather than lesser risk to the species. For example, we coded a species as 'likely' rather than 'insufficient information'. However, we respected the qualification of coded BRU threats (for example, as 'possible') (Extended Data Fig. 5). This also applied if there were contradictions between different pieces of information and data. We considered information in assessments to be current, recognizing that some assessments are older than 10 years (Supplementary Methods 2.3). Where threat codes were qualified (for example, 'past (unlikely to return)') we interpreted them as past or current accordingly (Supplementary Methods 2.4). Regarding flora, we treated species as threatened by use even if only 5.3.5 (BRU → Logging & wood harvesting → Motivation Unknown/Unrecorded) was coded as a threat unless it was evident in the text fields that the species was not a tree, it was stated that code 5.3.5 applies to the species' habitat (not to the species) or other information meant it was not relevant (for example, past threat). Following coding, species categorized as 'likely' and 'insufficient information' were checked for accuracy of coding.

### Taxonomy alignment
Following the categorization of species, we determined which species are, and are not, included in the CITES appendices to determine those taxa currently subject to CITES trade measures. The full list of official species names from the CITES appendices was downloaded from the Checklist of CITES Species[43] and cross-checked with all 21,715 species to determine if the names corresponded to CITES-listed species. We considered species to be the same when the scientific name matched, even though we acknowledge that the species concept may differ, as taxonomies differ between the Red List and CITES (Supplementary Methods 2.7). Where no match was found, synonyms were considered to ensure that species treated as synonyms by either IUCN or CITES, and which were accepted names in the other taxonomy, were

not overlooked. For potential matches involving synonyms, particularly cases involving two synonyms, additional verification was carried out by manually checking the Red List assessment to ensure that the match was logical; species too distantly related or clearly referring to a separate species were discounted. Higher taxonomic listings in CITES (for example, primates) were cross-checked to ensure that even where there was not an exact match in nomenclature, species on the Red List within the corresponding genus, family or order of relevance received the corresponding CITES listing. For example, if there was a newly described primate on the Red List not yet recognized in the CITES nomenclature, the species was assumed to be covered by the Appendix II listing for primates or the Appendix I listing for the relevant genus or family.

For species with CITES listings that only cover certain populations (for example, *Diospyros* populations of Madagascar) or involve other exclusions (for example, the *Euphorbia* listing only applies to succulents), the distribution or other attributes were checked, where feasible, to ensure that the CITES listing or characterization as 'non-CITES' was correct. Where uncertain, we consulted the CITES nomenclature specialists for fauna and flora, respectively.

### Species verification
As the Red List had been updated on completion of coding, we verified whether each species in our dataset remained distinct. We did this by cross-referencing the unique identifier number for all species on the 2020-1 version of the Red List and the species in our dataset to identify those species no longer on the Red List (for example, because their taxonomy had changed). The Species Information Service database, which is used to store all current and historic Red List assessment data, was used for this purpose. Following the removal of 25 species and addition of 74 species but removing 19 LC and DD species, resulted in 21,745 candidate species. We coded the additional species and cross-referenced them with the CITES listing information as described. A total of 3,815 of these species mapped to CITES-listed species.

### Species calculations
We calculated the number of species in each category (for example, likely), those included and excluded from CITES and the proportion of species with BRU as a threat that are likely to be threatened by international trade, or not, overall and by class. To account for the uncertainty of species categorized as having insufficient information we followed previous studies[44,45] to estimate the proportion of these species that would be expected to be categorized as likely and unlikely if there was sufficient information (Supplementary Methods 2.5). To compare species likely to be threatened by international trade with those taxa considered to be threatened by use and/or trade at the subinternational level according to the Red List, we calculated the difference between those species categorized as likely in our dataset and those with BRU threat codes for which there is no evidence that exploitation for international trade is a threat to the species. We did this overall and by class.

### Repeatability
We assessed if the process could be fully automated using an advanced automated coding method and used Fleiss' Kappa to test for agreement between approaches (Supplementary Methods 2.8). We retrospectively recoded all Actinopterygii ($n = 1,187$) and Amphibia ($n = 329$) species that were manually coded and compared the advanced and manual coding results. We also tested the advanced coding against the initial coding of all Actinopterygii and Amphibia species (that is, including taxa that were coded using the simpler automated coding) and tested whether it could correctly categorize species in these classes with new or updated assessments. We then tested the approach on all animals (kingdom Animalia) in our dataset. The advanced coding achieved 83% accuracy for Actinopterygii and Amphibia species compared to manual coding ($\kappa = 0.72$, 95% CI 0.68–0.75, $P = 0.000$) and 92% across

all species initially coded in these classes ($\kappa$ = 0.82, 95% CI 0.80–0.85, $P$ = 0.000) respectively. It achieved 88% accuracy for new or updated assessments. For all animals, it achieved 77% accuracy ($\kappa$ = 0.6, 95% CI 0.58–0.62, $P$ = 0.000). These results demonstrate that the advanced coding performs well (Supplementary Tables 3–6 and Supplementary Results 4.3) and this process can be used to generate data to inform decision-making in CITES (Supplementary Discussion 5.2).

### Reporting summary

Further information on research design is available in the Nature Portfolio Reporting Summary linked to this article.

## Data availability

Data generated in this study are included in the Supplementary Data.

## Code availability

Source code for advanced automated coding of species is available on GitHub (https://github.com/AlyPavitt/Challender.etal_IntTradeThreat).

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

## Acknowledgements

We thank P. P. van Dijk and R. Klopper for advice on nomenclature and C. Pollock for comments. D.W.S.C., K.M. and A.T.P. acknowledge funding from the UK Research and Innovation's Global Challenges Research Fund through the Trade, Development and the Environment Hub project (ES/S008160/1). C.H.-T. and J.S. acknowledge Environment Agency Abu Dhabi and Rufford Foundation for financial support. The views expressed in this paper do not necessarily reflect those of the IUCN.

## Author contributions

Conceptualization was by D.W.S.C., P.J.C., K.M., A.T.P., J.E.R., C.H.-T., J.S., R.K.B.J., R.H. and M.H. Methodology was developed by D.W.S.C., P.J.C., A.J., K.M., A.T.P., J.E.R., C.H.-T., J.S., R.K.B.J., R.H., T.E.E.O., R.H. and M.H. Queries and coding were done by D.W.S.C., P.J.C., A.J., K.M., A.T.P., J.E.R. and J.S. Analysis was undertaken by D.W.S.C., K.M. and A.T.P. The original draft was written by D.W.S.C., K.M., C.H.-T., R.K.B.J., T.E.E.O. and M.H. The final article was edited and reviewed by D.W.S.C., P.J.C., K.M., A.T.P, J.E.R., C.H.-T., J.S., R.H., A.J., T.E.E.O., R.K.B.J., D.A.C. and M.H.

## Competing interests

The authors declare no competing interests.

## Additional information

**Extended data** is available for this paper at https://doi.org/10.1038/s41559-023-02115-8.

**Correspondence and requests for materials** should be addressed to Daniel W. S. Challender.

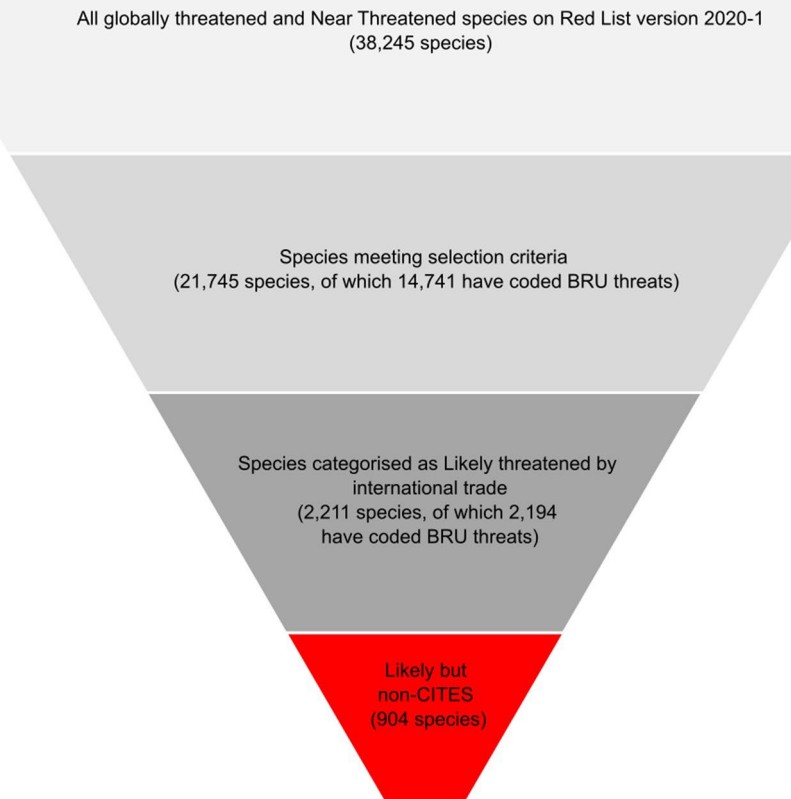

**Extended Data Fig. 1 | Filtering process.** Key data filtering steps to determine species likely threatened by international trade on the Red List and of those species which are, and are not, included in CITES.

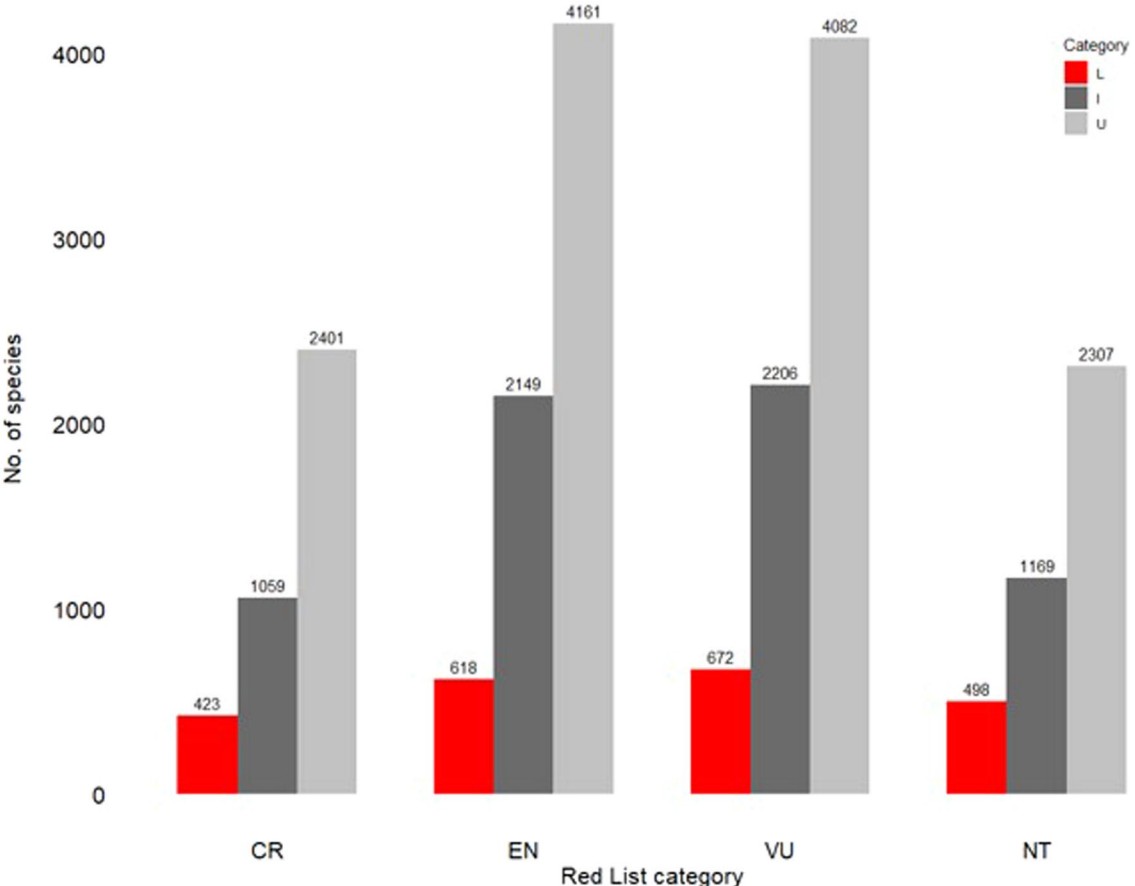

**Extended Data Fig. 2 | Number of species in categories Likely (L), Insufficient Information (I) and Unlikely (U) by Red List Category.** CR = Critically Endangered, EN = Endangered, VU = Vulnerable, NT = Near Threatened.

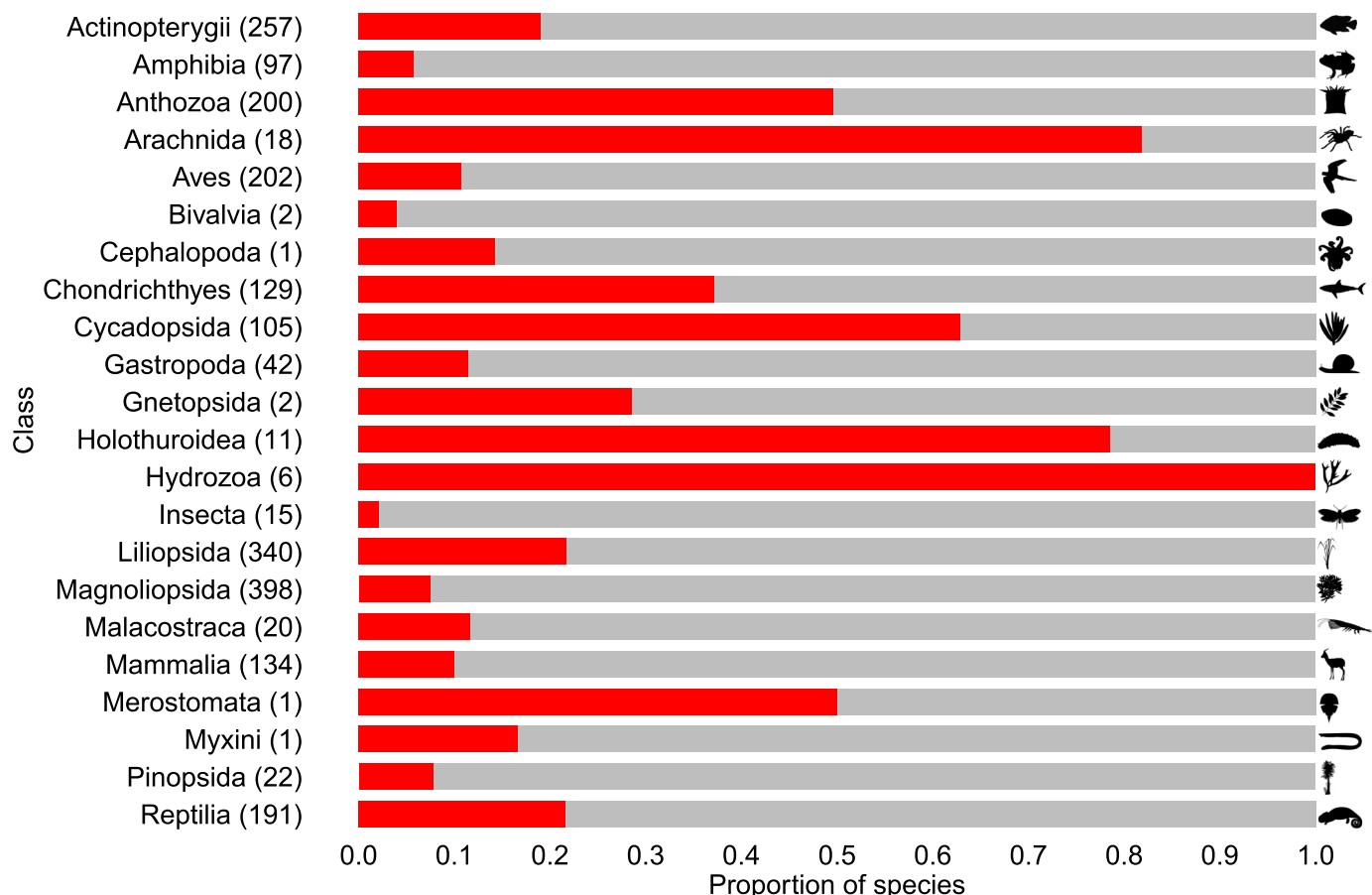

**Extended Data Fig. 3 | Threatened and Near Threatened species categorized as likely threatened by international trade (red) as a proportion of species in these Red List categories with intentional BRU threats coded (grey), by class. Credit for *Gazella gazella* image: Rebecca Groom, under a Creative Commons license CC BY 3.0 (without changes).** Numbers in parentheses are species likely threatened by international trade. Excludes 118 species likely threatened by international trade that do not have BRU threats coded (17 species) or intentional BRU threats coded (101 species).

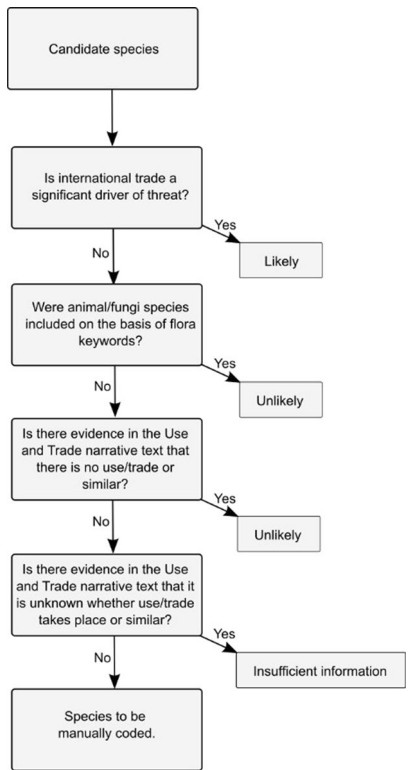

**Extended Data Fig. 4 | Decision tree for automated coding of species.** Initial automated coding was based on information in the 'use and trade' field, data on 'is international trade a significant driver of threat', and use-related threat codes.

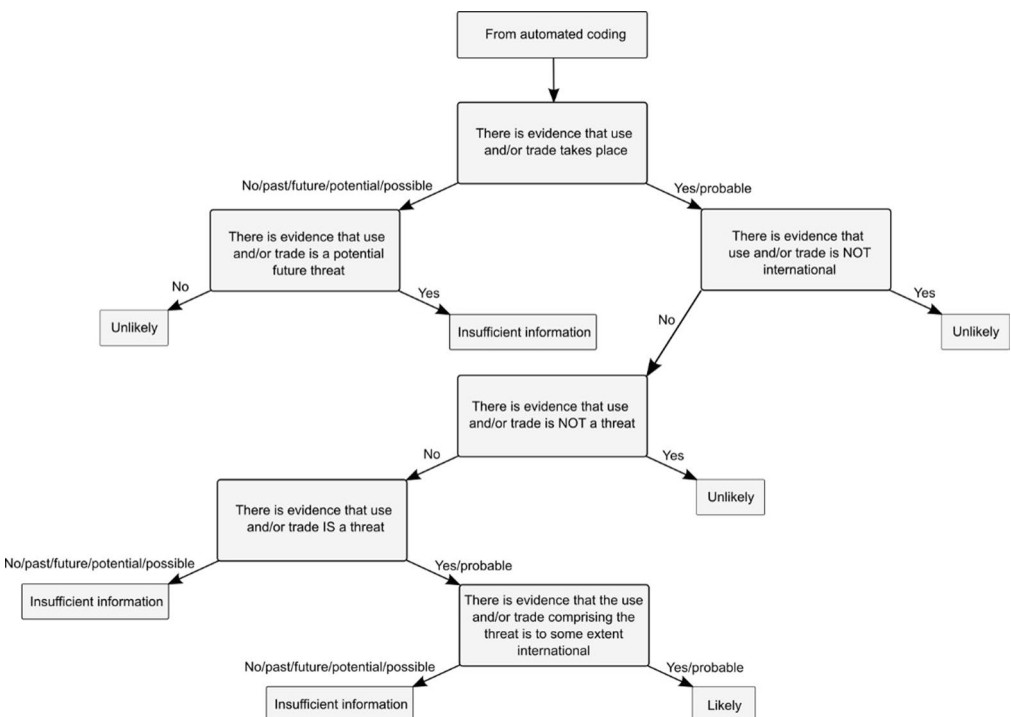

**Extended Data Fig. 5 | Decision tree for manual coding of species.** Manual coding entailed a person reading the information and data for each assessment and categorizing species aided by the decision tree. The information and data used in decision-making comprised the rationale, threats, and use and trade fields; threat codes; scale-of use codes; purpose of use codes; the 'no use/trade information on this species' field; and 'is international trade a significant driver of threat' field.

**Extended Data Table 1 | Number of threatened and Near Threatened species in categories Likely, Insufficient information and Unlikely**

| Class | Total species | Category | | |
|---|---|---|---|---|
| | | Likely | Insufficient information | Unlikely |
| Actinopterygii | 2007 | 260 | 632 | 1115 |
| Agaricomycetes | 65 | | 3 | 62 |
| Amphibia | 1640 | 97 | 88 | 1455 |
| Andreaeopsida | 1 | | | 1 |
| Anthocerotopsida | 1 | | | 1 |
| Anthozoa | 404 | 200 | 199 | 5 |
| Arachnida | 42 | 18 | 1 | 23 |
| Arthoniomycetes | 1 | | | 1 |
| Aves | 1423 | 202 | 335 | 886 |
| Bivalvia | 101 | 2 | 15 | 84 |
| Branchiopoda | 2 | | | 2 |
| Bryopsida | 81 | | 17 | 64 |
| Cephalaspidomorphi | 1 | | | 1 |
| Cephalopoda | 7 | 1 | 1 | 5 |
| Chondrichthyes | 317 | 129 | 114 | 74 |
| Clitellata | 34 | | 32 | 2 |
| Cycadopsida | 225 | 106 | 108 | 11 |
| Diplopoda | 55 | | | 55 |
| Enopla | 1 | | | 1 |
| Entognatha | 1 | | | 1 |
| Florideophyceae | 9 | | | 9 |
| Gastropoda | 1078 | 43 | 148 | 887 |
| Ginkgoopsida | 1 | | 1 | |
| Gnetopsida | 10 | 2 | 4 | 4 |
| Holothuroidea | 16 | 11 | 4 | 1 |
| Hydrozoa | 6 | 6 | | |
| Insecta | 881 | 15 | 103 | 763 |
| Jungermanniopsida | 35 | | 2 | 33 |
| Lecanoromycetes | 12 | | 6 | 6 |
| Leotiomycetes | 1 | | | 1 |
| Liliopsida | 2150 | 343 | 839 | 968 |
| Lycopodiopsida | 17 | | 2 | 15 |
| Magnoliopsida | 7745 | 402 | 3267 | 4076 |
| Malacostraca | 299 | 20 | 95 | 184 |
| Mammalia | 1189 | 134 | 169 | 886 |
| Marchantiopsida | 3 | | | 3 |
| Merostomata | 2 | 1 | 1 | |
| Myxini | 8 | 1 | 4 | 3 |
| Onychophora | 4 | | 2 | 2 |
| Pezizomycetes | 2 | | | 2 |
| Phaeophyceae | 5 | | | 5 |
| Pinopsida | 254 | 22 | 72 | 160 |
| Polypodiopsida | 126 | | 29 | 97 |
| Polytrichopsida | 1 | | | 1 |
| Reptilia | 1477 | 196 | 289 | 992 |
| Sarcopterygii | 2 | | 1 | 1 |
| Sordariomycetes | 1 | | | 1 |
| Sphagnopsida | 2 | | | 2 |
| TOTAL | 21,745 | 2,211 | 6,583 | 12,951 |

**Extended Data Table 2 | Number of species in each class categorized as Likely threatened by international trade included in the three CITES Appendices and not included in CITES**

| Class | Species Likely threatened by international trade | CITES status | | | | |
|---|---|---|---|---|---|---|
| | | Appendix I | Appendix II | Appendix I/II | Appendix III | Non-CITES |
| Actinopterygii | 260 | 3 | 26 | | | 231 |
| Amphibia | 97 | 2 | 37 | | 3 | 55 |
| Anthozoa | 200 | | 200 | | | |
| Arachnida | 18 | | 14 | | | 4 |
| Aves | 202 | 49 | 85 | | 5 | 63 |
| Bivalvia | 2 | | | | | 2 |
| Cephalopoda | 1 | | | | | 1 |
| Chondrichthyes | 129 | 3 | 37 | | | 89 |
| Cycadopsida | 106 | 53 | 53 | | | |
| Gastropoda | 43 | | 1 | | | 42 |
| Gnetopsida | 2 | | | | | 2 |
| Holothuroidea | 11 | | 3 | | 1 | 7 |
| Hydrozoa | 6 | | 6 | | | |
| Insecta | 15 | 1 | 10 | | | 4 |
| Liliopsida | 343 | 100 | 147 | | | 96 |
| Magnoliopsida | 402 | 33 | 181 | | | 188 |
| Malacostraca | 20 | | | | | 20 |
| Mammalia | 134 | 74 | 38 | 4 | 6 | 20 |
| Merostomata | 1 | | | | | 1 |
| Myxini | 1 | | | | | 1 |
| Pinopsida | 22 | 1 | 4 | | | 17 |
| Reptilia | 196 | 29 | 97 | 1 | 10 | 61 |
| TOTAL | 2,211 | 343 | 934 | 5 | 25 | 904 |

**Extended Data Table 3 | Number and Red List Category of species in each class categorized as Likely threatened by international trade and included in CITES**

| Class | Total species | Red List Category | | | |
|---|---|---|---|---|---|
| | | Critically Endangered | Endangered | Vulnerable | Near Threatened |
| Actinopterygii | 29 | 13 | 3 | 12 | 1 |
| Amphibia | 42 | 4 | 14 | 19 | 5 |
| Anthozoa | 200 | | 8 | 92 | 100 |
| Arachnida | 14 | 2 | 6 | 2 | 4 |
| Aves | 139 | 18 | 34 | 43 | 44 |
| Chondrichthyes | 40 | 20 | 10 | 9 | 1 |
| Cycadopsida | 106 | 27 | 35 | 28 | 16 |
| Gastropoda | 1 | | | | 1 |
| Holothuroidea | 4 | | 3 | 1 | |
| Hydrozoa | 6 | 1 | 2 | 2 | 1 |
| Insecta | 11 | | 2 | 3 | 6 |
| Liliopsida | 247 | 95 | 104 | 35 | 13 |
| Magnoliopsida | 214 | 55 | 74 | 63 | 22 |
| Mammalia | 114 | 23 | 35 | 43 | 13 |
| Pinopsida | 5 | 1 | 3 | 1 | |
| Reptilia | 135 | 36 | 43 | 39 | 17 |
| TOTAL | 1307 | 295 | 376 | 392 | 244 |

**Extended Data Table 4 | Number and Red List Category of species in each class categorized as Likely threatened by international trade but not included in CITES**

| Class | Total species | Red List Category | | | |
|---|---|---|---|---|---|
| | | Critically Endangered | Endangered | Vulnerable | Near Threatened |
| Actinopterygii | 231 | 25 | 56 | 68 | 82 |
| Amphibia | 55 | 6 | 19 | 15 | 15 |
| Arachnida | 4 | | 1 | 2 | 1 |
| Aves | 63 | 6 | 7 | 25 | 25 |
| Bivalvia | 2 | 1 | 1 | | |
| Cephalopoda | 1 | | | 1 | |
| Chondrichthyes | 89 | 7 | 18 | 29 | 35 |
| Gastropoda | 42 | 6 | 17 | 8 | 11 |
| Gnetopsida | 2 | | | | 2 |
| Holothuroidea | 7 | | 4 | 3 | |
| Insecta | 4 | | 3 | | 1 |
| Liliopsida | 96 | 30 | 26 | 26 | 14 |
| Magnoliopsida | 188 | 27 | 57 | 66 | 38 |
| Malacostraca | 20 | 11 | 7 | 2 | |
| Mammalia | 20 | | 7 | 6 | 7 |
| Merostomata | 1 | | | 1 | |
| Myxini | 1 | | | | 1 |
| Pinopsida | 17 | 1 | 2 | 7 | 7 |
| Reptilia | 61 | 8 | 17 | 21 | 15 |
| TOTAL | 904 | 128 | 242 | 280 | 254 |

# Reporting Summary

## Statistics

For all statistical analyses, confirm that the following items are present in the figure legend, table legend, main text, or Methods section.

| n/a | Confirmed | |
|---|---|---|
| ☐ | ☒ | The exact sample size (*n*) for each experimental group/condition, given as a discrete number and unit of measurement |
| ☐ | ☒ | A statement on whether measurements were taken from distinct samples or whether the same sample was measured repeatedly |
| ☐ | ☒ | The statistical test(s) used AND whether they are one- or two-sided <br> *Only common tests should be described solely by name; describe more complex techniques in the Methods section.* |
| ☒ | ☐ | A description of all covariates tested |
| ☐ | ☒ | A description of any assumptions or corrections, such as tests of normality and adjustment for multiple comparisons |
| ☐ | ☒ | A full description of the statistical parameters including central tendency (e.g. means) or other basic estimates (e.g. regression coefficient) AND variation (e.g. standard deviation) or associated estimates of uncertainty (e.g. confidence intervals) |
| ☐ | ☒ | For null hypothesis testing, the test statistic (e.g. *F*, *t*, *r*) with confidence intervals, effect sizes, degrees of freedom and *P* value noted <br> *Give P values as exact values whenever suitable.* |
| ☒ | ☐ | For Bayesian analysis, information on the choice of priors and Markov chain Monte Carlo settings |
| ☒ | ☐ | For hierarchical and complex designs, identification of the appropriate level for tests and full reporting of outcomes |
| ☒ | ☐ | Estimates of effect sizes (e.g. Cohen's *d*, Pearson's *r*), indicating how they were calculated |

*Our web collection on statistics for biologists contains articles on many of the points above.*

## Software and code

Policy information about availability of computer code

| Data collection | R (Version 4.0.3) was used to collect data from the IUCN Red List of Threatened Species and categorize species using automated coding and advanced automating coding approaches described in the article. Code for advanced automated coding of species is available from GitHub (link in article). |
|---|---|
| Data analysis | Custom code in R (Version 4.0.3) was used to categorize species using automated coding and advanced automating coding as described in the article. This entailed searching for particular keywords and text strings in selected text fields, and assessing the relevance of use-related threat codes, in IUCN Red List assessments. Code for advanced automated coding of species is available from GitHub (link in article). SPSS v.28 was used to estimate Fleiss' Kappa to compare manual, automated and advanced automated coding approaches. |

For manuscripts utilizing custom algorithms or software that are central to the research but not yet described in published literature, software must be made available to editors and reviewers. We strongly encourage code deposition in a community repository (e.g. GitHub). See the Nature Portfolio guidelines for submitting code & software for further information.

## Data

Policy information about availability of data

All manuscripts must include a data availability statement. This statement should provide the following information, where applicable:

- Accession codes, unique identifiers, or web links for publicly available datasets
- A description of any restrictions on data availability
- For clinical datasets or third party data, please ensure that the statement adheres to our policy

*Provide your data availability statement here.*

# Field-specific reporting

Please select the one below that is the best fit for your research. If you are not sure, read the appropriate sections before making your selection.

☐ Life sciences     ☐ Behavioural & social sciences     ☒ Ecological, evolutionary & environmental sciences

For a reference copy of the document with all sections, see nature.com/documents/nr-reporting-summary-flat.pdf

# Ecological, evolutionary & environmental sciences study design

All studies must disclose on these points even when the disclosure is negative.

| | |
|---|---|
| Study description | We queried the IUCN Red List of Threatened Species (version 2020-1) to identify species potentially threatened by international trade. Using selection criteria we then categorized 21,745 species as being 'Likely' or 'Unlikely' to be threatened by international trade, or as having 'Insufficient information' to determine the likelihood of this threat based on available information in IUCN Red List assessments. We subsequently determined which of the species we evaluated to be 'Likely' threatened by international trade are, and are not, included in the CITES Appendices, which included aligning taxonomies between the IUCN Red List and CITES. We discuss the results in the context of all threats to species from biological resource use (BRU) on the IUCN Red List, including the scale of threat: local, national, or international. |
| Research sample | We used data on the 38,245 threatened and Near Threatened species on the IUCN Red List of Threatened Species (version 2020-1) (Methods and Supplementary Methods 2.1-2.3). |
| Sampling strategy | We queried the IUCN Red List of Threatened Species (version 2020-1) to identify species potentially threatened by international trade using defined criteria (Methods and Supplementary Methods 2.1-2.3). |
| Data collection | A.J. undertook initial queries on the IUCN Red List of Threatened Species (version 2020-1) and shared the results, which were in .csv files, with D.W.S.C. Details of the queries conducted are in Methods and Supplementary Methods 2.1-2.3. D.W.S.C. cross-referenced query outputs and constructed a MS Excel database including assessment data for all species for subsequent automated and manual coding. K.M. aligned taxonomies between the IUCN Red List of Threatened Species and CITES and stored the results in a MS Excel file. These results were shared with D.W.S.C. for analysis. A.T.P. conducted the automated coding and advanced automated coding and stored the outputs in .csv files that were shared with D.W.S.C. for analysis. |
| Timing and spatial scale | Data from the IUCN Red List (version 2020-1) were initially downloaded in May 2020 and used for the initial coding of species, which was completed in February 2021. Data were further collected as needed (e.g., for the advanced automated coding) up until, and including, September 2021. |
| Data exclusions | No data were excluded from the analyses. |
| Reproducibility | Data and code are available. |
| Randomization | Not relevant. No randomization needed. |
| Blinding | Blinding was not relevant to this study. Results relating to the categorization of species were unobtainable until manual coding had been completed independently by D.W.S.C., J.E.R., P.J.C. and K.M. and subsequently centralized, and until K.M. had completed taxonomy alignment between the IUCN Red List and CITES. |

Did the study involve field work?     ☐ Yes     ☒ No

# Reporting for specific materials, systems and methods

We require information from authors about some types of materials, experimental systems and methods used in many studies. Here, indicate whether each material, system or method listed is relevant to your study. If you are not sure if a list item applies to your research, read the appropriate section before selecting a response.

## Materials & experimental systems

| n/a | Involved in the study |
|---|---|
| ☒ | ☐ Antibodies |
| ☒ | ☐ Eukaryotic cell lines |
| ☒ | ☐ Palaeontology and archaeology |
| ☒ | ☐ Animals and other organisms |
| ☒ | ☐ Human research participants |
| ☒ | ☐ Clinical data |
| ☒ | ☐ Dual use research of concern |

## Methods

| n/a | Involved in the study |
|---|---|
| ☒ | ☐ ChIP-seq |
| ☒ | ☐ Flow cytometry |
| ☒ | ☐ MRI-based neuroimaging |

