## [Peer Review File · Nature Ecology & Evolution]

Peer Review Information

Journal: Nature Ecology & Evolution

Manuscript Title: Identifying species likely threatened by international trade on the IUCN Red List can inform CITES trade measures

Corresponding author name(s): Daniel W.S. CHALLENGER

Editorial Notes:

Redactions – transferred manuscripts (mention of previous referee reports from elsewhere)

This manuscript has been previously reviewed at another journal. This document only contains reviewer comments, rebuttal and decision letters for versions considered at Nature Ecology & Evolution. Mentions of prior referee reports have been redacted

Redactions – transferred manuscripts (mention of the other journal)

This manuscript has been previously reviewed at another journal. This document only contains reviewer comments, rebuttal and decision letters for versions considered at Nature Ecology & Evolution. Mentions of the other journal have been redacted.

Reviewer Comments & Decisions:

Decision Letter, first revision:

1st November 2022

Dear Dr Challender,

Your manuscript entitled "Identifying species likely threatened by international trade on the IUCN Red List can inform CITES trade measures" has now been seen by 3 reviewers, whose comments are attached. I'd like to express my sincere thanks for your patience while we gathered these reviews; I am very sorry that it has taken this long, but I am glad we waited.

[REDACTED]

Having carefully considered these three reviews and the previous reviews, we are prepared to further consider the paper for publication in NEE. The reviewers have raised a number of concerns which will need to be addressed before we can offer publication in Nature Ecology & Evolution. We will therefore need to see your responses to these concerns, along with a revised manuscript, before we can reach a final decision regarding publication.

We therefore invite you to revise your manuscript taking into account all reviewer comments. Please highlight all changes in the manuscript text file.

* If you have not done so already please begin to revise your manuscript so that it conforms to our Article format instructions at <http://www.nature.com/natecolevol/info/final-submission>. Refer also to any guidelines provided in this letter.

2* Include a revised version of any required reporting checklist. It will be available to referees (and, potentially, statisticians) to aid in their evaluation if the manuscript goes back for peer review. A revised checklist is essential for re-review of the paper.

[REDACTED]

Nature Ecology & Evolution is committed to improving transparency in authorship. As part of our efforts in this direction, we are now requesting that all authors identified as 'corresponding author' on published papers create and link their Open Researcher and Contributor Identifier (ORCID) with their account on the Manuscript Tracking System (MTS), prior to acceptance. ORCID helps the scientific community achieve unambiguous attribution of all scholarly contributions. You can create and link your ORCID from the home page of the MTS by clicking on 'Modify my Springer Nature account'. For more information please visit www.springernature.com/orcid.

[REDACTED]

Reviewers' comments:

Reviewer #1 (Remarks to the Author):

Challender et al creates an assessment tool that uses the IUCN Redlist data to generate an assessment of threat associated with trading a species as the international scale. The derive three criteria - Likely, Insufficient, and Unlikely - across a broad range of taxa and show that over 14,000 species are being used and/or traded from which ~6% are Likely threatened by international trade. The authors further explore these data in context to the CITES listings and show that 59% are listed in CITES Appendices and the remaining 900 species are currently omitted from the Appendices. Thus authors highlight the need to revisit these 900 species owing to their high risk from the trade.

2Moreover, the intent of the study appear to be less focused (or perhaps equate parts) on the empirical findings but rather emphasizing the need for a systematic approach to assess trade risk that can be utilized by CITES parties for real-time assessment.

I enjoyed reading this paper and I agree with the need for a tool that can autonomously generate risk assessments of trade over time. Such integration between IUCN and CITES has been asked for in numerous recent studies and it appears that this approach taken by Challender et al is an initial step to accomplish this goal. That said, I do have some reservations with the manuscript and its approach in its current state.

My major concerns that I would like the authors to address are as follows:

1) I think there needs to be some further clarification and additional analyses that takes into account national/domestic pressures from trade. I see this as a critical issue for the utility of this risk assessment tool and how the assessment tool might be used to inform CITES decision making. It is the intent that this assessment tool be used by managers and policy makers (as stated in line 199, fig 4, etc) and as such it requires a full appreciate of all levels of trade with associated levels of certainty and confidence. As an example – your assessment criteria is heavily weighted on the risk posed by international trade (this is showcased in the Insufficient information category methods and I do recognize that it is the core intent of your paper) but by doing this you may have a species that is severely threatened by domestic/national trade but ranked as Insufficient or Unlikely for international trade. International trade cannot be decoupled from national trade as they both take from the same wild population. As such, to me it makes little sense to ignore national level pressures when determining the Likely threat of trade. But if national trade is so severe as to threaten the species wouldn't this be revealed in a correctly executed NDF by the exporting country when a species is being considered for international trade? Importantly, might your assessment be misused as evidence indicating Unlikely or Insufficient risk despite there being considerable risk to the species if it were approved for trade (owing to national level pressures)? Does it in this scenario further complicate decision making rather than improving it? I feel rather strongly that the authors provide a very convincing response to this concern. My preference (and apologies for suggesting additional work) would be to expand the risk assessment criteria to include a bare minimum indicator of national level threats so that species with Insufficient or Unlikely risk will minimally be flagged and thus shift the onus to a thorough critique of the NDF. I do note that I am aware that in Step 2b use and trade at the local and/or national level is considered and so I am asking for this information to carry-forward as a weighting on your downstream risk criteria for international trade. In some ways this provides checks and balances within your assessment tool. We are unable to predict the widespread adoption of this tool but what happens if your tool is welcomed by CITES and it is someday weighted considerably or equally to the NDF? It may be totally adopted and so some care and critical thought should be given to how it could be misused (equally to that of its intended use). The authors even acknowledge this important point on line 333-336 where national threats are present in some ~70% of species.

2) Following the point above, it appears that a national assessment was indeed already performed (e.g., Lines 57 and 332), yet, I have scoured the manuscript several times and I cannot find any methods or results reported on this. If you already completed the national assessment then wonderful as this make addressing my comment above much easier. My recommendation is to report the number of species in a venn diagram or some other method for Likely/Insufficient/Unlikely for

3international trade against Likely/Insufficient/Unlikely for national trade. This will help understand those that are Insufficient/Unlikely threatened by international trade that may be Likely threatened by national trade thus warranting more caution by CITES for these select species. Perhaps even something similar to Supp Table 6?

3) Additionally, it would be good to also weight the three criteria by the time since last Redlist assessment. The authors have put forth a procedure for detecting the risk but time since assessment would then provide some estimation of confidence behind this. E.g., I would have much more confidence in an Unlikely score for a species assessed in 2020 than a species assessed in 2010. It appears the authors have these data (line 653). This is a standard approach in other disciplines such as climate change where detection/attribution is married with confidence. I do note that the authors are aware of the importance of certainty and confidence denoted by their approach in line 460 of the supp material.

4) Why exclude Least Concern species at this point? Isn't the purpose of this study to assess risk and thus if it is traded, then the species should be assessed. To do otherwise, simply suggests that you might use IUCN Redlist criteria as the sole factor determining trade risk. In other words, you have to include everything even if the end result is that these species end up as Unlikely traded. I am okay with excluding DD species, but I really think LC species need to be considered as part of this assessment. I think these data are also useful to establish a baseline and trends over time

5) I appreciate the decision trees outlined in extended data figure 4 and 5. I strongly recommended that you try to incorporate additional information here to provide even further clarity of the decision process. For example, where on the decision tree do BRUs come into play? In other words, indicate which parts of the Redlist data are being used at each step. This is an extended figure and so it doesn't need to look beautiful. I think these additional details will make it abundantly clear how and when/where the data are being used; E.g., describe the 'evidence' for each step. Please note that it took me some time of reading and rereading to grasp the data flow and decision making and I'm still not 100% certain I understand all the nuances of your approach. These decision trees, when expanded, can solve this problem.

There are some confusing areas throughout the methods that need attention in order to fully understand what was done. Importantly, some areas are confusing which prevented my understanding of whether it affected the assessment criteria or not.

line 119 – although I appreciate the need for brevity in journals like NEE I really think it is important to include some indication of what these selection criteria are within the main body of the manuscript – preferably here at line 119. Does this mean simply traded vs not traded?

Line 174

This is a perplexing statement. If you scored a 59% on a test would you say that you did 'moderately well'? To me a C on a test (70-79%) is moderate performance. Really? Why are you inclined to give CITES so much credit?

This part of the sentence seems to be the most important part in my opinion:

"especially considering the lack of a readily accessible evidence base to date on species threatened by overexploitation for international trade."

Line 194

4Length of time – see my comment below on including this in further analysis. I think this is an important consideration to build confidence around your criteria.

Line 224

Please note here that when talking about threat categories the entire discussion is contingent on the assessment being reasonably up-to-date. I really believe your tool would benefit from a confidence metric that includes time since the completion of the redlist assessment. I'm certain the authors are well aware of how quickly market demand can shift in the trade and so a species NT from 10 years ago may very well be in worse condition today, etc.

Line 231

Clarify what is meant by fragmented

Line 240-242

By incorporating the date since assessment as an indicator of confidence one can easily use this tool for its intended purposes while having a clear understanding for its limitations. This could also help IUCN prioritize assessments (I am not familiar with how this decision is made)? E.g., if a Likely traded species hasn't been assessed in the past 10 years OR if you have a CR species with Insufficient information

line 318-320

This sentence is wordy and a bit difficult to understand. I'm not sure it is grammatically correct. Either way, please reword to add clarity.

Line 325

This is very confusing. Where in the manuscript do you state that you assessed national level trade? Where is this coming from? If you have done this then wonderful; see my comment on adding this as an additional metric.

Line 332

I don't understand why a midpoint approach is being used (actually I do not quite understand what the approach is at all; midpoint of what?) when you have actual data on international and national.

Methods

Line 483 – why exclude Least Concern species at this point? Isn't the purpose of this study to assess risk and thus if it is traded then the species should be assessed. To do otherwise, simply suggests that you can use IUCN Redlist criteria as the sole factor determining trade risk. I am okay with excluding DD species, but I really think LC species need to be considered as part of this assessment.

Line 526

I find Query 3 to be quite confusing. So was it used or not? Saying that it isn't reliable doesn't sound convincing? I do think it is important information but in Query 1 and 2 you report the final number of species whereas here it remains unclear how many species resulted from the query as well as the general purpose of the query. If the point is to keep this query into the framework because it should improve over time then be sure to say this.

Line 536

Again, same as above. Provide the number of species from the query. Does Q#4 yield 5 species or 5000?

Line 554

Please insert the Query #s next to each scoring criteria so that it is more obvious how Q1-5 filter into these threat assignments.

Line 604

It is confusing how this falls under the precautionary approach: "even if it is a species known to be impacted by international trade from other information sources, then it was categorised as Insufficient information or Unlikely based on the information available."

Line 614

How were these species selected? Do you mean the 9320 species were the only species to have data from the "the "use and trade" and "is international trade a significant driver of threat" fields, and the relevance of use-related threat codes"

Line 617 -625

I'm not understanding this section on Automated coding. It sounds like you are applying a new criteria for Likely, Insufficient and Unlikely. This is very confusing and if true, then more information needs to be provided as to why this is being done. My assumption would be that you are doing this to test the efficiency of an automated system but nowhere in this section is that stated. It is briefly mentioned on line 551 but then why redefine the criteria? It seems like what ever criteria being made starting on line 554 – 598 should be equally applied regardless if it is a human or computer.

Line 623-625

How is this automated? Sounds like a person is making this decision e.g., timber?

Line 649

Information here is hard to interpret. What does this mean? i.e., what does it mean that you 'respected' the BRU code? "However, we respected the qualification of coded BRU threats (e.g., as "possible") (Extended Data Fig. 5)." Does this mean you removed species or shifted them to across criteria? If yes, then way more detail is needed here with a full report of its effect on the sample sizes.

Line 656

I recommend providing the acronym in the text above; yes a bit redundant but with the methods at the end a reader not familiar with this acronym will have to go find its meaning which is unnecessarily inconvenient.

Reviewer #2 (Remarks to the Author):

6The idea that IUCN should play a greater pre-emptive role in CITES proposals and reviews could be a reasonable one, however IUCN data on trade and threats is rarely an accurate representation of a species' current situation, and many species evaluated by the IUCN do not list trade – despite some/many being heavily traded. Because the IUCN Red List is an unreliable source of data on trade, this study does not contribute much that hasn't been conducted before.

Without international trade monitoring of species not listed in CITES, it can be very difficult for IUCN experts to gauge the true level of species use. This can even be an issue for CITES-listed species, e.g., the top US imported reptile; *Testudo horsfieldii*, had its last IUCN assessment in 1996 (vulnerable) and lists no trade threat or presence in international trade despite >200k live individuals being imported in the past decade

Relying on IUCN data to determine which species are most at risk from trade is only going to be useful if the data in question is accurate and up to date, which is not satisfactorily addressed by the authors.

I don't think the Red List provides sufficient information to determine which species would benefit from future CITES listing. The Red List itself estimates that only 10% of all species have been assessed ([https://www.iucnredlist.org/about/faqs#Why is the species I am looking for not on the Red List](https://www.iucnredlist.org/about/faqs#Why%20is%20the%20species%20I%20am%20looking%20for%20not%20on%20the%20Red%20List)), with around 26% of terrestrial vertebrates and 36% of reptiles not assessed (<https://doi.org/10.1016/j.biocon.2019.108203>).

In terms of previous studies comparing between CITES and the IUCN Red List, a recently published study (<https://doi.org/10.5204/ijcjsd.1945>) examined species threatened by trade, their presence in CITES appendices and the Red List, and some factors that might influence their listing in CITES. They also explored taxonomic biases in CITES and the IUCN.

In addition, (<https://doi.org/10.1126/science.aav4013>) looked at species threatened by international trade, their presence in CITES appendices and the Red List, and the time difference between a listing in CITES and an assessment by the IUCN, for species which included harvesting as a 'threat', albeit not species in 'use and trade'.

Another study (<https://doi.org/10.1126/science.aav5327>) examined trade in terrestrial vertebrates and included a section on the difference in species listed in CITES and traded according to the Red List, using very similar methods to the current study.

Reviewer #3 (Remarks to the Author):

I share the sentiments of one previous reviewer in that I am excited about what this paper sets out to do, i.e. to develop and test a method for extracting information from IUCN Red List assessments that can increase the scientific evidence base for CITES listing decisions. The expressed intention to support a more systematic approach to listing species on CITES is certainly needed.

The authors have given detailed answers to previous reviewers and have addressed all of the questions relating to methods and interpretation in the amended manuscript. I don't have any further comment on methods but I have the following comments on interpretation and analysis.

7line 88 non-detriment finding (typo = funding)

Line 115. I am not convinced by the more ambitious statement that “we provide the first systematic assessment of the threat posed by international trade across all taxonomic groups” (114-116). This should refer to the ‘potential threat posed by international trade’ -the paper provides and tests a method to systematically identify species potentially impacted by international trade based on information captured in the IUCN Red List database. The actual assessment of the threat posed by international trade is constrained by the IUCN Red List data (as the authors acknowledge in the SOM discussion, and see comments later in this review).

Line 192. Its not clear why anthozoans, hydrozoans and arachnids are singled out as higher level taxa that are Likely threatened by international trade and listed on CITES Appendices. Unless this is meant to represent a particular taxonomic level, the same is true for orchids, cycads and pangolins.

Line 257-300. There is a long and speculative discussion of issues that CITES needs to consider when listing species. The cautionary notes regarding CITES are all valid because CITES is limited to responses that regulate or restrict trade, when these may not be the best policy options under all circumstances. However, these arguments detract from the main findings of this study and the authors do not present any mechanism by which further analysis of IUCN Red Lists or the associated automated processes can resolve these issues. The discussion therefore digresses into speculative territory (CITES could do this or could do that). It should suffice to say that the scientific evidence from the analysis of the IUCN Red List can identify species that may be at risk from international trade and these species should be further evaluated by CITES. When Parties consider whether or not to list the species on the CITES Appendices, they should take other factors into consideration to determine if such a listing will ultimately lead to better outcomes for the species. Although this is seldom done by CITES, the authors can reference other papers that have highlighted the need for CITES to consider a wider range of factors when making listing decisions (including those by the authors) (e.g. Challender et al 2015, Cooney et al 2021, IPBES 2022).

Lines 212- Instead of the more speculative options available to CITES it would be more robust and helpful to emphasize additional steps for further analysis of the IUCN data and to determine whether these can also be automated. This is important because for many species, the IUCN data and expertise is the best available and CITES will rely on the same data or experts to justify listing proposals. One obvious outcome is flagging higher level taxonomic groups where CITES may have a blind spot. This is dealt with in text (Line 216 and Fig. 3) but could be emphasized as an important part of the process in Fig 4. Ray-finned fish are a good example but even for birds and Amphibians, are there particular families that are being overlooked or is it just a random sample across families. Many groups have been overlooked by CITES. Sometimes this is deliberate (e.g. certain fish species because they are supposed to be governed by other treaties, or sharks and rays because regulation first had to overcome logistical difficulties with enforcement) but can also reflect the biases highlighted in the paper. The process should flag these high level gaps and allow CITES to examine the reasons for such gaps.

Further steps associated with the analysis of IUCN Red List data will be needed to support an evaluation by CITES. The authors have taken a precautionary approach to identifying species that are

8Likely affected by international trade. I support this approach but it can lead to a misinterpretation of gaps, unless this is specified in the main paper, and should be used to guide next steps for CITES. Previous reviewers have interpreted the 41% of “Likely” species that are not listed on CITES as a massive failure of CITES, suggesting that the findings of the study were an indictment of CITES. There are certainly many shortcomings in CITES processes but there are several reasons why a ‘Likely’ finding from the IUCN database may not indicate a failure on the part of CITES. It is important to frame this correctly in the main paper so that the paper (and the underlying methods) have maximum value and impact for CITES. Some of these reasons are outlined in the SOM but should be given greater prominence in the main text. For example, I took a random sample of six CR and EN species coded as Likely to be impacted by international trade, one (*Aetobatus flagellum*) stands out as a species that likely should be covered by CITES. The others (*Phytotriades auratus*, *Crax pinima*, *Dypsisceracea*, *Agave warreliana*, *Serranochromis robustus*) all refer to international trade but provide either: (i) less compelling evidence of any impact; (ii) lack of clarity on international versus domestic trade; or (iii) segmented impacts where the international trade component has a low impact. This implies that the real gaps in CITES listings may be far less than 41%. I don’t want the authors to waste time interrogating these examples- the point I want to make is that the next steps for CITES can be informed by additional information in the IUCN Red List database. These steps should be further articulated in relation to Fig.4 and the authors should confirm whether any of these steps could be automated or, to apply the cautionary principle, further analysis of Red List data is best done manually as part of the CITES review process.

•

*****END*****

Author Rebuttal, first revision:

Challender et al. – Identifying species likely threatened by international trade on the IUCN Red List can inform CITES trade measures

Reviewer comment (left hand column) and responses (right hand column).

Reviewer #1 (Remarks to the Author)	
Challender et al creates an assessment tool that uses the IUCN Redlist data to generate an assessment of threat associated with trading a	Thank you for this succinct summary of our research and we are glad you agree with the premise of the study.

9species as the international scale. The derive three criteria - Likely, Insufficient, and Unlikely - across a broad range of taxa and show that over 14,000 species are being used and/or traded from which ~6% are Likely threatened by international trade. The authors further explore these data in context to the CITES listings and show that 59% are listed in CITES Appendices and the remaining 900 species are currently omitted from the Appendices. Thus authors highlight the need to revisit these 900 species owing to their high risk from the trade. Moreover, the intent of the study appear to be less focused (or perhaps equate parts) on the empirical findings but rather emphasizing the need for a systematic approach to assess trade risk that can be utilized by CITES parties for real-time assessment. I enjoyed reading this paper and I agree with the need for a tool that can autonomously generate risk assessments of trade over time. Such integration between IUCN and CITES has been asked for in numerous recent studies and it appears that this approach taken by Challender et al is an initial step to accomplish this goal. That said, I do have some reservations with the manuscript and its approach in its current state.	
My major concerns that I would like the authors to address are as follows: 1) I think there needs to be some further	Thank you for this suggestion, which is appreciated. We agree that providing some insight on species which our

clarification and additional analyses that takes into account national/domestic pressures from trade. I see this as a critical issue for the utility of this risk assessment tool and how the assessment tool might be used to inform CITES decision making. It is the intent that this assessment tool be used by managers and policy makers (as stated in line 199, fig 4, etc) and as such it requires a full appreciate of all levels of trade with associated levels of certainty and confidence. As an example – your assessment criteria is heavily weighted on the risk posed by international trade (this is showcased in the Insufficient information category methods and I do recognize that it is the core intent of your paper) but by doing this you may have a species that is severely threatened by domestic/national trade but ranked as Insufficient or Unlikely for international trade. International trade cannot be decoupled from national trade as they both take from the same wild population. As such, to me it makes little sense to ignore national level pressures when determining the Likely threat of trade. But if national trade is so severe as to threaten the species wouldn't this be revealed in a correctly executed NDF by the exporting country when a species is being considered for international trade? Importantly, might your assessment be misused as evidence indicating Unlikely or Insufficient risk despite there being considerable risk to the species if it were approved for trade (owing to national level pressures)? Does it in this scenario further complicate decision making rather than

analyses - current and future - indicate are not threatened by international trade but rather use/trade at the sub-international level would be helpful. However, precisely because the core intent of the analyses and paper was to understand the number of, and which, species are threatened by international trade (as per the purview of the Convention), or not, our methods have been tailored to this task, including our categories and criteria, our decision trees, and software code. As such, it will be difficult for us to provide comparative assessment of the threat to species from local and/or domestic use and/or trade – to the point where we can categorise species as Likely, Unlikely, and Insufficient information for these scales of trade - without fundamentally redesigning the study, including the above methodological components. Moreover, beyond the Red List, we are not aware of a credible and comprehensive data source that can be relied upon to support such an analysis consistently for the many thousands of species we evaluate herein. That said, what is possible based on our results, and as we note below this may be what the Reviewer was asking after anyway, is to provide an output which includes all species that have biological resource use (BRU) threat codes applied which our analyses indicate are not threatened by

improving it? I feel rather strongly that the authors provide a very convincing response to this concern. My preference (and apologies for suggesting additional work) would be to expand the risk assessment criteria to include a bare minimum indicator of national level threats so that species with Insufficient or Unlikely risk will minimally be flagged and thus shift the onus to a thorough critique of the NDF. I do note that I am aware that in Step 2b use and trade at the local and/or national level is considered and so I am asking for this information to carry-forward as a weighting on your downstream risk criteria for international trade. In some ways this provides checks and balances within your assessment tool. We are unable to predict the widespread adoption of this tool but what happens if your tool is welcomed by CITES and it is someday weighted considerably or equally to the NDF? It may be totally adopted and so some care and critical thought should be given to how it could be misused (equally to that of its intended use). The authors even acknowledge this important point on line 333-336 where national threats are present in some ~70% of species.

international trade but rather by use and/or trade at the sub-international level (which we hasten to add is itself a key result of our paper, sensu the penultimate sentence of our abstract). This would be a list of species not threatened by international trade as per the figures for different classes in Fig. 2. This could be provided along with future outputs of species that are ‘Likely’ threatened by international trade. Going forward this would focus on species for which assessments are new and/or updated. It would enable Parties to use this list of species – which have been filtered through our analyses such that it is known that the threat from overexploitation is at the sub-international level – and examine them further, including scrutiny of NDFs to be made on a case by case basis. We hope that the reviewer would be satisfied with our proposed solution as a way to address the need they have outlined and in a way which would be useful for the Parties. We have included such an output in Supplementary Data 2 based on the current dataset and referred to this in the article. This data file also includes the current Appendix on which species are listed and otherwise indicates whether species are not currently included in CITES. Text at Lines 366-369 now reads as follows:

“Finally, future iterations of our

analyses could explicitly indicate CITES-listed species that are threatened by local and/or domestic use and/or trade rather than exploitation for international trade, and be shared with the CITES Parties to inform appropriate actions, including scrutiny of NDFs (Supplementary Data 2).”

Regarding misuse of evidence, we are not convinced that our results would be misused as evidence. We envision a scenario where we share future iterations of our results with the CITES Parties at the most appropriate time in the intersessional period between CoPs, most likely with the Animals and Plants Committees in their first meeting after a CoP. We are currently discussing how this could work with the Animals Committee Chair. Decisions were also adopted at CoP19 (see here and here) which provide a mandate for the Parties to examine what such a process may look like. On sharing these results we would make it abundantly clear to the Parties (and observers) what the results do and do not indicate, i.e., species that have been categorised as ‘Likely’ threatened by international trade have been so because they meet our criteria, which would also be provided in our outputs. Similarly, we would highlight that species categorised as ‘Unlikely’ and

	‘Insufficient Information’ have been evaluated for whether there is threat from international trade and the categorisation of species as such does not indicate that they are not threatened by use and/or trade at the sub-international level. Providing the additional output referred to above would allow the Parties to examine and scrutinize this list of species, including when considering making NDFs on a case by case basis. As articulated in the manuscript and in response to past and current reviewer comments, we would expect the Parties to also draw on other, additional information and data sources (e.g., peer-reviewed literature) to inform any decision-making – both in relation to potential proposals to amend the Appendices and any other uses including informing NDFs – which would hopefully also reduce any chance of our results being misused as evidence.
2) Following the point above, it appears that a national assessment was indeed already performed (e.g., Lines 57 and 332), yet, I have scoured the manuscript several times and I cannot find any methods or results reported on this. If you already completed the national assessment then wonderful as this make addressing my comment above much easier. My recommendation is to report the number of species in a venn diagram or some other method for Likely/Insufficient/Unlikely for international trade against	Thank you for raising this point. We do want our methods and results to be as clear as possible. We have included further detail on the approach we took at Lines 130-134. This text now reads as follows: “We then identified which of these species are, and are not, included in the CITES Appendices and evaluated the results in the context of threats to species from biological resource use (BRU),

Likely/Insufficient/Unlikely for national trade. This will help understand those that are Insufficient/Unlikely threatened by international trade that may be Likely threatened by national trade thus warranting more caution by CITES for these select species. Perhaps even something similar to Supp Table 6?	including comparing species “Likely” threatened by international trade with those considered threatened by use and/or trade at the local and/or domestic level by the IUCN Red List..” We have also included further detail on the methods used to calculate the # of species Likely threatened by international trade and those threatened by use and/or trade at the local and/or domestic level in the ‘Species calculations’ part of the Methods. Text at Lines 737-742 now reads: “To compare species Likely threatened by international trade with those taxa threatened by use and/or trade at the sub-international level, we calculated the difference between those species categorised as Likely in our dataset and those with BRU threat codes for which there is no evidence that exploitation for international trade is a threat to the species. We did this overall and by class.” The results of this analyses are included in Fig. 2, in the section titled Threats in Context (Lines 339-383), and the list of species threatened by local and/or domestic use and/or trade is included in Supplementary Data 2.
--	--

3) Additionally, it would be good to also weight the three criteria by the time since last Redlist assessment. The authors have put forth a procedure for detecting the risk but time since assessment would then provide some estimation of confidence behind this. E.g., I would have much more confidence in an Unlikely score for a species assessed in 2020 than a species assessed in 2010. It appears the authors have these data (line 653). This is a standard approach in other disciplines such as climate change where detection/attribution is married with confidence. I do note that the authors are aware of the importance of certainty and confidence denoted by their approach in line 460 of the supp material.	Thank you for this suggestion. We discussed this at length during our study formulation, and with colleagues. We haven't built time since last assessment into our criteria (which we elaborate on a bit further below where the reviewer also raises this point), but we are mindful of the need to consider it in our results. Our suggested approach here is to include "time since last assessment" or "date of last assessment" as an explicit column in any tabulated output that would be shared with the CITES Parties in the future. This would enable the Parties to consider quickly and easily how recent the information contained in a particular Red List assessment is, and to interpret the result of our analysis in that light. Both in reference to potential proposals to amend the Appendices and NDFs we would expect, and indeed we strongly urge, the Parties (and other organisations) to consult other information sources as well to inform their decision-making, as we have stated clearly in the manuscript.
4) Why exclude Least Concern species at this point? Isn't the purpose of this study to assess risk and thus if it is traded, then the species should be assessed. To do otherwise, simply suggests that you might use IUCN Redlist criteria as the sole factor determining trade risk. In other words, you have to include everything even if the end result is that these species end up as Unlikely traded. I am okay	Thank you for this suggestion. When developing the study design and the methods we did explicitly consider whether or not to include LC species. We ultimately felt that our primary focus should be on threatened and NT species, because these species are those most likely to be in potential need of trade measures in CITES where they are not

with excluding DD species, but I really think LC species need to be considered as part of this assessment. I think these data are also useful to establish a baseline and trends over time.	currently in place and also most likely to meet the biological criteria. We do appreciate the suggestion but our preference is to retain the focus on threatened and NT species.
5) I appreciate the decision trees outlined in extended data figure 4 and 5. I strongly recommended that you try to incorporate additional information here to provide even further clarity of the decision process. For example, where on the decision tree do BRUs come into play? In other words, indicate which parts of the Redlist data are being used at each step. This is an extended figure and so it doesn't need to look beautiful. I think these additional details will make it abundantly clear how and when/where the data are being used; E.g., describe the 'evidence' for each step. Please note that it took me some time of reading and rereading to grasp the data flow and decision making and I'm still not 100% certain I understand all the nuances of your approach. These decision trees, when expanded, can solve this problem.	Thank you for raising this. We do want our methods and results to be as clear as possible and we appreciate there are many nuances in analysing Red List data in a rigorous and defensible way. We explain how we went about assigning species to a category at Lines 628-639 and 641-691 in the manuscript for both the automated and manual categorisation of species. And the reader can examine the criteria at Lines 575-626. To improve the clarity of our decision making process, we have included further detail on the fields used in the categorisation process in the figure captions, which we hope addresses this point. EDF4 caption now reads: “Extended Data Fig. 4 Decision tree for automated coding of species. Initial automated coding was based on information in the “use and trade” field, data on “is international trade a significant driver of threat”, and use-related threats codes.” EDF5 caption now reads: “Extended

	Data Fig. 5 Decision tree for manual coding of species. Manual coding entailed a human being reading the information and data for each assessment and categorising species aided by the decision tree. The information and data used in decision making comprised the rationale, threats, and use and trade fields; threat codes; scale of use codes; purpose of use codes; the “no use/trade information on this species” field; and “is international trade a significant driver of threat” field.”
There are some confusing areas throughout the methods that need attention in order to fully understand what was done. Importantly, some areas are confusing which prevented my understanding of whether it affected the assessment criteria or not. ????	Thank you for raising these points. We address them directly below.
line 119 – although I appreciate the need for brevity in journals like NEE I really think it is important to include some indication of what these selection criteria are within the main body of the manuscript – preferably here at line 119. Does this mean simply traded vs not traded?	We are happy to comply, and to leave the journal editor to make a decision if this presents a problem in terms of length. We have included some additional detail on the selection criteria which we hope addresses this point. The text at Lines 120-127 now reads as follows: “Starting with >38,000 globally threatened and Near Threatened species that are categorised as globally threatened—Critically Endangered, Endangered or Vulnerable—and Near Threatened on the Red List (version

	2020-1), we used selection criteria to identify species potentially threatened by international trade. The selection criteria comprised threatened (species that are categorised as Critically Endangered, Endangered, or Vulnerable) and Near Threatened threat categories, relevant threat codes, the presence of particular terms within assessments (e.g., “commercial use”), and information on the scale of end uses for species (e.g., “subsistence” or “international”).”
Line 174 This is a perplexing statement. If you scored a 59% on a test would you say that you did ‘moderately well’? To me a C on a test (70-79%) is moderate performance. Really? Why are you inclined to give CITES so much credit? This part of the sentence seems to be the most important part in my opinion: “especially considering the lack of a readily accessible evidence base to date on species threatened by overexploitation for international trade.”	We included this sentence only because we think it is helpful in providing context to the results. Against a backdrop of no readily accessible database on species threatened by overexploitation for international trade (to use the reviewer’s analogy – not having studied for the test), we consider the inclusion of 59% of species that we categorised as “Likely” to be threatened by international trade on the Red List to be moderate success for CITES in terms of including such species in the Appendices. Were such a database to be available, one could rightly criticise the CITES Parties more pointedly for not including a higher proportion of such species because they would have had an accessible evidence base on which to draw. We appreciate that this assessment is somewhat subjective but we hope that we have demonstrated through this review process that we have thought

	about this and evaluated it carefully, including paying close attention to the language used and the rationale for qualifying success in the way we have. That said, we are happy to reconsider this wording if the reviewer feels strongly otherwise.
Line 194 Length of time – see my comment below on including this in further analysis. I think this is an important consideration to build confidence around your criteria.	We address this point directly where it is raised above and in the point below but provide some detail here for convenience. Our suggested approach is to include “time since last assessment” or “date of last assessment” as a column in any tabulated output that would be shared with the CITES Parties in the future. This would enable the Parties to consider quickly and easily how recent the information contained in particular Red List assessments is. We (and CITES) would expect the Parties to consider other information in decision-making as well though as articulated in responses to other points raised by the reviewers.
Line 224 Please note here that when talking about threat categories the entire discussion is contingent on the assessment being reasonably up-to-date. I really believe your tool would benefit from a confidence metric that includes time since the completion of the redlist assessment. I’m certain the authors are well aware of how quickly market demand can shift in the trade	We have three points of response. The first is that among comprehensively assessed species on the Red List for which documentation of use is available, only a small proportion (i.e., 6%) of species have assessments that need updating. Thus, we don’t consider that many assessments will be affected in the way that the reviewer is suggesting and indeed previous analyses (e.g. Marsh et

and so a species NT from 10 years ago may very well be in worse condition today, etc.	al. 2022; Cons Biol) have demonstrated this. However, even if they were, our second point is that our analyses are only one source of information that we (and CITES) would expect the Parties and other organisations to draw on when making decisions on whether a CITES listing proposal should be submitted, as we believe we have made clear in the paper. Third, we do agree that including information on time since last assessment would be useful for the CITES Parties (by way of a confidence metric, as suggested by the reviewer) and as such, we would include such data in any outputs provided to the Parties in the future, e.g., by indicating “time of last assessment” or similar.
Line 231 Clarify what is meant by fragmented	We are making the point that the current process of Parties submitting proposals to amend the Appendices is not systematic. We acknowledge fragmented was too cryptic and have now changed this to “unsystematic”.
Line 240-242 By incorporating the date since assessment as an indicator of confidence one can easily use this tool for its intended purposes while having a clear understanding for its limitations. This could also help IUCN prioritize assessments (I am not familiar with how this decision is made)? E.g., if a Likely traded species hasn't been assessed in the past 10 years OR if you	We agree that we could solve the timing issue the Reviewer raises by including information on “time since last assessment” or similar when sharing future iterations of the results with the CITES Parties. Doing so would allow us to highlight, and the Parties to recognise, that where Red List assessments were conducted some time ago there may well be new or additional information that is

have a CR species with Insufficient information	not contained in the assessments but which may be useful to informing decision-making.
line 318-320 This sentence is wordy and a bit difficult to understand. I'm not sure it is grammatically correct. Either way, please reword to add clarity.	We have edited the sentence – now at Lines 340-343 – and it now reads: “Ensuring that species threatened by international trade are identified and , where they would be likely to benefit in conservation terms, international trade controls established, where such species would be likely to benefit in conservation terms, is a crucial step to safeguarding species from overexploitation.”
Line 325 This is very confusing. Where in the manuscript do you state that you assessed national level trade? Where is this coming from? If you have done this then wonderful; see my comment on adding this as an additional metric.	Thank you for raising this point. We have now included further text on this in the paragraph before the Results and Discussion section. Lines 130-134 now read as follows: “We then identified which of these species are, and are not, included in the CITES Appendices and evaluated the results in the context of threats to species from biological resource use (BRU), including comparing species “Likely” threatened by international trade with those considered threatened by use and/or trade at the local and/or domestic level by the IUCN Red List.” We hope this makes this clearer, although we are happy to make further amendments to ensure this is readily understandable by the reader.

Line 332

I don't understand why a midpoint approach is being used (actually I do not quite understand what the approach is at all; midpoint of what?) when you have actual data on international and national.

Further explanation here may help. We categorised c.22,000 species into three categories 'Likely', 'Unlikely' and 'Insufficient information'. An obvious next question is how many of the 'Insufficient information' species would be 'Likely' threatened by international trade if there were sufficient information. Using a method used in many previous studies for accounting for the uncertainty introduced by Data Deficient species in reporting % of species threatened (noting that DD species might be threatened, but might not; see, for example, references 44 and 45), we estimated the number of species categorised as 'Insufficient Information' that would be 'Likely' threatened by international trade, overall and by class, if sufficient information were available. We present three values in the paper i) lower-bound, which assumes that none of the species coded 'Insufficient information' are 'Likely', ii) mid-point, which assumes that species coded 'Insufficient information' have the same fraction of species coded 'Likely' as among species coded 'Likely' and 'Unlikely' and, iii) upper-bound, which assumes that all of the species coded 'Insufficient information' are 'Likely'. This method allowed us to account for uncertainty associated with species categorised as 'Insufficient information'. We explain this in Supplementary Methods 2.5 and present results in

	Supplementary Results 4.1 and 4.2. We discuss the results regarding species threatened by local and/or domestic use and/or trade and those threatened by international trade at Lines 339-369.
Methods Line 483 – why exclude Least Concern species at this point? Isn't the purpose of this study to assess risk and thus if it is traded then the species should be assessed. To do otherwise, simply suggests that you can use IUCN Redlist criteria as the sole factor determining trade risk. I am okay with excluding DD species, but I really think LC species need to be considered as part of this assessment.	We address this under #4 above but provide a response here as well for convenience. When developing the study design and the methods we did explicitly consider whether or not to include LC species. We ultimately felt that our initial focus should be on threatened and NT species because these species are those most likely to meet the criteria in CITES Resolution conf 9.24 (Rev CoP17) (given how these define threatened species). Following the CITES CoP19 meeting we will shortly be discussing this research, the results, and the proposed process with the CITES Parties. While we feel justified in retaining the focus on threatened and NT species only at this stage, we are amenable to including LC species in future iterations of our results if this is something that the CITES Parties would find useful and helpful.
Line 526 I find Query 3 to be quite confusing. So was it used or not? Saying that it isn't reliable doesn't sound convincing? I do think it is important	There is some nuance here and it might be helpful for us to explain it. For a subset of threat codes related to intentional use (5.1.1, 5.2.1, 5.3.1, 5.3.2, 5.4.1, and 5.4.2) assessors are asked to

information but in Query 1 and 2 you report the final number of species whereas here it remains unclear how many species resulted from the query as well as the general purpose of the query. If the point is to keep this query into the framework because it should improve over time then be sure to say this.

indicate whether international trade is a significant driver of threat to species – with possible responses comprising yes/no/unknown. This functionality was added to the Red List database only recently and as such it has only been used in a subset of assessments to date. Had this functionality been built into the Red List database at its inception it would have been used in many more assessments. This field is also not consistently applied. As we explain in Lines 554-562 in the manuscript, this field is therefore not a reliable indicator of species for which international trade is a significant driver of threat. However, importantly, where this field has been used and assessors have indicated that international trade is a significant driver of threat to species, these data can be used to categorise species as ‘Likely’ as per our categories and criteria (see Lines 575-626 in the Methods. As we note at Lines 636–639 we did not use other responses (‘no’ and ‘unknown’) because the aim of the study was to determine whether international trade posed any level of threat to species rather than being a significant driver of threat necessarily.

This query did not generate additional taxa for our candidate list of species, but it did provide information on

	international trade being a significant driver of threat for 50 species in our list of candidate species (i.e., for these species, assessors had used this field to indicate that international trade is a significant driver of threat). These data were used in the categorisation of species as per our categories and criteria (Lines 575-626 in the Methods). As additional taxa were not generated with this query we did not include mention of the number of species concerned to avoid confusing the reader. We have sought to clarify how this Red List field was used by adding text to the methods section. Lines 561-562 now reads as follows: “However, as data from this field can indicate whether international trade is a significant driver of threat for a subset of species, we included these data to aid categorisation of species.”
Line 536 Again, same as above. Provide the number of species from the query. Does Q#4 yield 5 species or 5000?	We thank the reviewer for this suggestion but we are concerned that including the number of species this query relates to may cause further confusion. This is because not all queries generated new species to go on our candidate list of species. Like query #3, this query simply generated additional data for species already on the list, data that were helpful for categorising

	species. Regarding query 4, it was data on all coded threats to species on the Red List. We have reviewed the methods section on queries and do think it is clear regarding which queries generated species for consideration and which queries generated additional data to consider in the categorisation of species.
Line 554 Please insert the Query #s next to each scoring criteria so that it is more obvious how Q1-5 filter into these threat assignments.	Thank you to the reviewer for this suggestion but we are not convinced that including query # next to the categories and criteria would help elucidate how we categorised species. As we explain at Lines 658-691, manual coding entailed a human being using the MS Excel database that we compiled using the query results and aided by the decision tree in EDF5, categorising species accordingly. We are also concerned that including query number would confuse the reader. We believe the methods section sets out the decision making process in sufficient detail but would be happy to make further edits to aid understanding.
Line 604 It is confusing how this falls under the precautionary approach: “even if it is a species known to be impacted by international trade from other information sources, then it was categorised as Insufficient information or Unlikely based on the information available.”	We can see why you raised this point and indeed it is fundamental to understanding the results and why we believe them to be conservative. Our precautionary approach is bounded by the information available in the Red List assessments and as per our categories and criteria and coding approach we assumed greater rather than lesser risk to species as

	detailed at Lines 677-678 in the methods. Importantly, by design we didn't refer to any information or our own intuition beyond the explicit documentation in the Red List assessments. This was to ensure the results can be replicated and repeated by others directly from the Red List.
Line 614 How were these species selected? Do you mean the 9320 species were the only species to have data from the “the “use and trade” and “is international trade a significant driver of threat” fields, and the relevance of use-related threat codes”	Our initial automated coding was used to assign a subset of species to one of three categories (Likely, Unlikely, Insufficient information), where these species could be easily categorised based on one or two data fields only (e.g., is international trade a significant driver of threat = yes, or the use and trade field included relevant information e.g., “no known human use”). The 9,320 species were the only species that fulfilled these criteria and hence could be categorised automatically using our initial automated coding. This is explained further in Supplementary Information section 2.4. The rest of the species (12,395 species) were manually coded. We then developed advanced automated coding to improve the repeatability of the methods. This is discussed at Lines 745-760 and in Supplementary Methods 2.8 and Supplementary Results 4.3.
Line 617 -625 I'm not understanding this section on Automated coding. It sounds like you are applying a new criteria for Likely, Insufficient	Thank you for this feedback. We are not applying new criteria here but just applying the existing criteria using an automated process where it was possible to categorise species using one or two

and Unlikely. This is very confusing and if true, then more information needs to be provided as to why this is being done. My assumption would be that you are doing this to test the efficiency of an automated system but nowhere in this section is that stated. It is briefly mentioned on line 551 but then why redefine the criteria? It seems like what ever criteria being made starting on line 554 – 598 should be equally applied regardless if it is a human or computer.	data fields only (see above).
Line 623-625 How is this automated? Sounds like a person is making this decision e.g., timber?	This was an automated process – and further details are provided in Supplementary Information 2.4 – but these results were then spot-checked by a human being. We have included mention of this checking at Lines 653-655: “These automation processes were tested extensively during development and were subsequently spot checked by a human being for accuracy.”
Line 649 Information here is hard to interpret. What does this mean? i.e., what does it mean that you ‘respected’ the BRU code? “However, we respected the qualification of coded BRU threats (e.g., as “possible”) (Extended Data Fig. 5).” Does this mean you removed species or shifted them to across criteria? If yes, then way more detail is needed here with a full report of its effect on the sample sizes.	Thank you for this feedback. We didn’t remove species or shift them across criteria but we respect (i.e. take into account) the qualification of the BRU threats. For instance, if a species was described as having a “potential” threat from international trade, then it would have been coded as “Insufficient information” rather than “Likely”, respecting the qualifying text provided in the Red List assessment.

	Included here is one of the examples from Supplementary Information 2.4 which might help demonstrate this. Aurelio’s rock lizard Iberolacerta aurelio was categorised as Insufficient information. The species has a relevant threat code applied (5.1.1) and the scale of use field indicates that the species is used internationally, which the end use field indicates is for research purposes. However, the threat from intentional use, described as “collection”, is qualified as being “possible” in the threats field. As such, using the decision tree in EDF5 this species was categorised as Insufficient information rather than Likely.
Line 656 I recommend providing the acronym in the text above; yes a bit redundant but with the methods at the end a reader not familiar with this acronym will have to go find its meaning which is unnecessarily inconvenient.	Thank you for this. We have checked the manuscript and BRU is defined on first mention outside the abstract on p.5 at Line 132. However, we have included it earlier on the paragraph you mention for the convenience of the reader. Thank you again for the careful review and helpful remarks.
Reviewer #2 (Remarks to the Author)	
The idea that IUCN should play a greater pre-emptive role in CITES proposals and reviews could be a reasonable one, however IUCN data on trade and threats is rarely an accurate	Thank you for this feedback and we are glad you agree with the initial premise of the paper. While it is correct that IUCN Red List assessments are not mandated

representation of a species' current situation, and many species evaluated by the IUCN do not list trade – despite some/many being heavily traded. Because the IUCN Red List is an unreliable source of data on trade, this study does not contribute much that hasn't been conducted before.	to include information on the use and trade of species, many assessments do and numerous scientific studies have been published over the last two decades (and longer) that have made use of such data to understand threats to species, including a focus on use and/or trade. The Red List includes assessments of extinction risk for >147,000 species and has a target of assessing a further 129,000 by 2030. We therefore expect our results to be increasingly robust in the future. We also note that we refer to caveats associated with using IUCN Red List data for the purposes we have done in the Supplementary Information file. We therefore consider that this is a useful systematic starting point to highlight species that could be considered in further detail (i.e. using additional information sources) against the CITES listing criteria. We also note that the CITES CoP19 meeting adopted Decisions, the implementation of which will include examining data from the Red List and how it can help inform potential proposals to amend the CITES Appendices (see here and here).
Without international trade monitoring of species not listed in CITES, it can be very difficult for IUCN experts to gauge the true level of species use. This can even be an issue for CITES-listed species, e.g., the top US imported reptile; Testudo horsfieldii, had its last IUCN assessment in 1996 (vulnerable) and	Thank you for this feedback. As per the point immediately above, when IUCN Red List assessments are conducted it is not mandatory for assessors to include information on use and trade of species, though it is encouraged. However, many assessments do in fact contain such

lists no trade threat or presence in international trade despite >200k live individuals being imported in the past decade Relying on IUCN data to determine which species are most at risk from trade is only going to be useful if the data in question is accurate and up to date, which is not satisfactorily addressed by the authors.	information and many studies have been published that have made use of such data. It is also worth noting that most assessments relating to species in use and trade (for comprehensively assessed taxa) are kept up to date. Among comprehensively assessed species on the Red List for which documentation of use is available, only a small proportion (i.e., 6%) of species have assessments that need updating. We do appreciate the reviewer’s point regarding the need to monitor trade in species not listed in CITES. Recent research has suggested that many non-CITES listed species may be at risk from overexploitation, including for international trade. We consider such research to complement the approach we propose. We consider that CITES Parties and other organisations would be better informed in their decision-making by considering our results and future iterations, in addition to other information and analyses, including that from recent peer-reviewed literature that has examined trade in non-CITES listed species.
I don’t think the Red List provides sufficient information to determine which species would benefit from future CITES listing. The Red List itself estimates that only 10% of all species	Thank you for this feedback. It will perhaps be useful if we clarify the approach we have taken in the manuscript. Recognising limitations and

have been assessed

(<https://www.iucnredlist.org/about/faqs#Why> is the species I am looking for not on the Red List), with around 26% of terrestrial vertebrates and 36% of reptiles not assessed (<https://doi.org/10.1016/j.biocon.2019.108203>).

caveats to the IUCN Red List in terms of taxonomic coverage – which we have set out in detail in the Supplementary Information – our study uses available information in c.22,000 Red List assessments to categorise species as ‘Likely’ or ‘Unlikely’ to be threatened by international trade or if there is ‘Insufficient information’ to be able to make this determination. We are suggesting that our results and future iterations will be helpful to the CITES Parties in thinking about potential future proposals to amend the Appendices. Our results do not, and are not designed to, determine which species would – and would not - actually benefit from future CITES listings. We are acutely aware of differences between the IUCN Red List and CITES in terms of evaluating extinction risk (see Challender et al. 2019 here and Challender et al. 2022 here) and have worded our manuscript carefully in this regard. We do emphasise in our manuscript the need for Parties in thinking about potential future proposals to amend the Appendices to explicitly consider additional data and information sources beyond our results in order to determine whether the relevant listing criteria are met for the species in question.

Finally, we note that the IUCN Red List

	has assessed more species than the reviewer suggests. For example, as detailed on this link, only 13% of reptiles have not been assessed.
In terms of previous studies comparing between CITES and the IUCN Red List, a recently published study (https://doi.org/10.5204/ijcjsd.1945) examined species threatened by trade, their presence in CITES appendices and the Red List, and some factors that might influence their listing in CITES. They also explored taxonomic biases in CITES and the IUCN.	Thank you for raising this point. We are aware of the study concerned and have discussed the approach taken in that paper with the lead author. We note that the study – like others we have discussed in the Supplementary Information – does not present the methods used in a systematic way. As we hope the reviewer acknowledges, using data from the IUCN Red List is not necessarily straightforward and requires an in-depth knowledge of the relevant fields in order to interpret the data accordingly. This is why our manuscript includes such a detailed methods section, including in the Supplementary Information. Also critical to the process we are proposing is being able to review information in the Red List assessments in a standardised way such that the process can be replicated consistently in future. In this regard, we measured the inter-rater reliability of human coders prior to undertaking the manual coding element of our full dataset of c.22,000 species and compared our “automated” and manual coding methods as discussed in the Repeatability section of our methods (Lines 745-760). All of this said, we have included discussion of the study raised by the reviewer in our

	Supplementary Context section (Lines 60-64) in the Supplementary Information. We do highlight that the study did not demonstrate a systematic approach: “Finally, a further study examined species in international trade that are “at-risk” on the IUCN Red List and which of these species are included and excluded from CITES but the authors presented little information on the methods used and did not demonstrate the application of a systematic approach.”
In addition, (https://doi.org/10.1126/science.aav4013) looked at species threatened by international trade, their presence in CITES appendices and the Red List, and the time difference between a listing in CITES and an assessment by the IUCN, for species which included harvesting as a ‘threat’, albeit not species in ‘use and trade’.	Thank you for raising this point. We are aware of this study and discuss key limitations to it in our Supplementary Context section in the Supplementary Information. Key limitations to the study include that it: i) focused on intentional uses in Red List assessments only, ii) overlooked important coded information in assessments related to use and trade, iii) overlooked whether the use/trade comprising the threat is international or not, and iv) the methods used to infer whether “international trade is a factor in the species’ endangerment” or not were not systematic.
Another study (https://doi.org/10.1126/science.aav5327) examined trade in terrestrial vertebrates and included a section on the difference in species	Thank you for raising this. We are aware of this study and refer to it in our Supplementary Context section in the Supplementary Information. A key

listed in CITES and traded according to the Red List, using very similar methods to the current study.	difference between our study and this one is that the latter focuses on terrestrial vertebrates only; we include all taxonomic groups in our study. Our study is therefore more comprehensive when determining the threat to species from international trade using the IUCN Red List. We also note that we have taken extreme care when designing our study to ensure that it is robust. Perhaps the key criticism of this paper is that the authors considered all species ‘in trade’ to be ‘threatened by trade’ which is not the case. This is discussed further in Challender et al. 2022 (here).
Reviewer #3 (Remarks to the Author)	
I share the sentiments of one previous reviewer in that I am excited about what this paper sets out to do, i.e. to develop and test a method for extracting information from IUCN Red List assessments that can increase the scientific evidence base for CITES listing decisions. The expressed intention to support a more systematic approach to listing species on CITES is certainly needed.	Thank you for the positive assessment of our research.
The authors have given detailed answers to previous reviewers and have addressed all of the questions relating to methods and interpretation in the amended manuscript. I don’t have any further comment on methods	Thank you for your review, which is appreciated.

but I have the following comments on interpretation and analysis.	
line 88 non-detriment finding (typo = funding)	Thank you. We have now amended this to say ‘finding’ as intended.
Line 115. I am not convinced by the more ambitious statement that “we provide the first systematic assessment of the threat posed by international trade across all taxonomic groups” (114-116). This should refer to the ‘potential threat posed by international trade’ - the paper provides and tests a method to systematically identify species potentially impacted by international trade based on information captured in the IUCN Red List database. The actual assessment of the threat posed by international trade is constrained by the IUCN Red List data (as the authors acknowledge in the SOM discussion, and see comments later in this review).	Thank you for raising this point. We do want to be clear on what the study is, and is not, and communicate this clearly. We have amended the wording so that this sentence now reads as follows at Lines 116-120: “Using The IUCN Red List of Threatened Species™9 (hereafter “Red List”), widely acknowledged as the most authoritative source on extinction risk and threats to species globally, we provide the first systematic assessment of the likelihood of threat posed by international trade across all taxonomic groups (Methods, Supplementary Context, Supplementary Methods 2.1–2.8).” Otherwise, we recognise there are limitations to this approach using the Red List – which you appropriately highlight here – and which we have paid careful attention to in terms of including appropriate caveats both within the main text and the Supplementary Information.
Line 192. Its not clear why anthozoans, hydrozoans and arachnids are singled out as higher level taxa that are Likely threatened by international trade and listed on CITES	Thanks for this. We included these selected groups to highlight that there are taxonomic classes in which all or most species categorised as ‘Likely’

Appendices. Unless this is meant to represent a particular taxonomic level, the same is true for orchids, cycads and pangolins.	threatened by international trade are included in the CITES Appendices. This sentence does not say that all or most of the species in these classes are included in CITES rather that all or most of the species in these classes that we categorised as being ‘Likely’ threatened by international trade are included in CITES as per Extended Data Table 2.
Line 257-300. There is a long and speculative discussion of issues that CITES needs to consider when listing species. The cautionary notes regarding CITES are all valid because CITES is limited to responses that regulate or restrict trade, when these may not be the best policy options under all circumstances. However, these arguments detract from the main findings of this study and the authors do not present any mechanism by which further analysis of IUCN Red Lists or the associated automated processes can resolve these issues. The discussion therefore digresses into speculative territory (CITES could do this or could do that). It should suffice to say that the scientific evidence from the analysis of the IUCN Red List can identify species that may be at risk from international trade and these species should be further evaluated by CITES. When Parties consider whether or not to list the species on the CITES Appendices, they should take other factors into consideration to determine if such a listing will ultimately lead	Thank you for this feedback. Our reason for including this section is because we thought it was important to highlight that just because the CITES Parties may decide to list species in the Appendices that does not inherently mean that this is the most appropriate policy option for species because of e.g., potential perverse impacts of these measures. We also included this section because previous reviewers asked us to consider more explicitly the limitations to CITES, which then takes the discussion to compliance mechanisms and options within CITES to improve implementation for particular taxa. As such, we highlight that there are ways to reduce uncertainty associated with CITES listings regarding likely conservation outcomes for species. This includes the Parties thinking more deeply and realistically about the impact of trade measures and how they may affect trade

to better outcomes for the species. Although this is seldom done by CITES, the authors can reference other papers that have highlighted the need for CITES to consider a wider range of factors when making listing decisions (including those by the authors) (e.g. Challender et al 2015, Cooney et al 2021, IPBES 2022).

systems (e.g., different actors, consumers, and harvest and trafficking incentives) in the real world. The Parties could do this by considering additional information when deliberating on proposed amendments to the Appendices. In doing so, we do refer to others that have published in this space (e.g., Cooney *et al.* 2021) while being mindful not to over self-cite or indeed reach too far beyond the original intent of this study. At Lines 298-318 we have sought to discuss potential further actions that could be taken for species once they have been included in the Appendices. Again, the thinking here was to include explicit discussion on this because once species are included in the Appendices the reality is that there are a range of measures that the Parties can take to support compliance regarding particular species and/or Parties. We consider this important because it demonstrates that we have thought about how our results may integrate into the working of the Convention. We have also been mindful to explicitly avoid saying anything like “include species in the Appendices and they shall be saved” because we know that this is not the case.

All of this said, we have edited the first of the paragraphs raised by the reviewer to remove arguably the most esoteric

section on particular methods that could be used by Parties to reduce uncertainty around listing decisions. The paragraph at Lines 274-296 now reads as follows:

“Where species are considered to meet the listing criteria, proposing Parties and/or the Animals/Plants Committees should explicitly evaluate whether the proposed measures would realistically be expected to contribute to the conservation of the species, or not, and any associated risks¹⁷. This is critically important because while it is difficult to predict the effectiveness of CITES trade measures, they may sometimes do more harm than good for species (e.g., by removing conservation incentives¹⁹ or lead to accelerated wild harvest of species²⁰). Parties should consider assessing how and why particular outcomes may be expected based on an understanding of the relevant SESs, including how harvest incentives may change, how actors along supply chains may respond, and any likely adverse impacts¹⁷. Uncertainty could be further reduced by Parties identifying additional measures that would be needed to mitigate any identified risks and support the implementation of trade measures. This could include, for example, greater resources for law enforcement agencies to ensure adequate probabilities of

apprehension for would-be offenders, the establishment of partnerships with local communities to sustainably manage species, and/or programmes to change consumer behaviour²². Where species would be likely to benefit from trade measures, Parties could submit proposals to the next CoP, and the scientific committees could also recommend that proposals be submitted to these meetings (Fig. 4). This process would complement the submission of proposals based on other (e.g., national and/or NGO) priorities.”

Regarding “resolving these issues”, we do note that we have other papers in preparation that address how to reduce such uncertainty further. One of these papers includes a social-ecological systems framework of international wildlife trade, the use of which could be used to understand particular trade systems with a view to identifying areas of concern regarding likely conservation outcomes for species linked to CITES trade controls and appropriate mitigation measures.

Finally, we appreciate the reviewer raising these points. We have considered them but if the feeling from the Reviewer remains that this this detracts from the

	paper then we could consider moving it to the supplementary information document.
Lines 212- Instead of the more speculative options available to CITES it would be more robust and helpful to emphasize additional steps for further analysis of the IUCN data and to determine whether these can also be automated. This is important because for many species, the IUCN data and expertise is the best available and CITES will rely on the same data or experts to justify listing proposals. One obvious outcome is flagging higher level taxonomic groups where CITES may have a blind spot. This is dealt with in text (Line 216 and Fig. 3) but could be emphasized as an important part of the process in Fig 4. Ray-finned fish are a good example but even for birds and Amphibians, are there particular families that are being overlooked or is it just a random sample across families. Many groups have been overlooked by CITES. Sometimes this is deliberate (e.g. certain fish species because they are supposed to be governed by other treaties, or sharks and rays because regulation first had to overcome logistical difficulties with enforcement) but can also reflect the biases highlighted in the paper. The process should flag these high level gaps and allow CITES to examine the reasons for such gaps.	Thank you for this feedback and the suggestion. While we have retained our (amended) discussion on options available to CITES Parties once species have been included in the Appendices (see responses above), we have taken on board this feedback and suggestion. Specifically, we have highlighted that repeat iterations of our research can be used to identify classes and families of species that may have been overlooked by CITES (for whatever reason) and that our results can help inform processes to address this. The text at L238-240 now reads as follows: “These results can inform deliberations on potential proposals to revise trade measures for species ahead of CITES CoPs and can highlight overlooked taxonomic groups that may warrant greater attention under the Convention.”
Further steps associated with the analysis of IUCN Red List data will be needed to support an evaluation by CITES. The authors have	Thank you for this feedback and the suggestion. We are glad you support the precautionary approach taken as it is

taken a precautionary approach to identifying species that are Likely affected by international trade. I support this approach but it can lead to a misinterpretation of gaps, unless this is specified in the main paper, and should be used to guide next steps for CITES. Previous reviewers have interpreted the 41% of “Likely” species that are not listed on CITES as a massive failure of CITES, suggesting that the findings of the study were an indictment of CITES. There are certainly many shortcomings in CITES processes but there are several reasons why a ‘Likely’ finding from the IUCN database may not indicate a failure on the part of CITES. It is important to frame this correctly in the main paper so that the paper (and the underlying methods) have maximum value and impact for CITES. Some of these reasons are outlined in the SOM but should be given greater prominence in the main text. For example, I took a random sample of six CR and EN species coded as Likely to be impacted by international trade, one (*Aetobatus flagellum*) stands out as a species that likely should be covered by CITES. The others (*Phytotriades auratus*, *Crax pinima*, *Dypsis ceracea*, *Agave warreliana*, *Serranochromis robustus*) all refer to international trade but provide either: (i) less compelling evidence of any impact; (ii) lack of clarity on international versus domestic trade; or (iii) segmented impacts where the international trade component has a low impact. This implies that the real gaps in CITES listings may be far less than 41%. I don’t want the authors to waste time

something we discussed at length at the beginning. We do agree that further analysis will be needed by the Parties based on our results. We envision a scenario where we share the results periodically with the Parties and the Parties and other organisations consider additional data and information (e.g., from peer-reviewed literature) to inform their decision-making regarding the potential submission of proposals to amend the Appendices and subsequently decision-making on listing proposals (by Parties only). The Parties may also wish to use the additional output suggested by Reviewer 1 to further scrutinize NDFs.

Regarding the misinterpretation of gaps, thank you for this feedback as well. We have now included further explanation of the results and how they should, and should not, be interpreted in the main text, focusing on species categorised as Likely threatened by international trade. Lines 329-332 now read as follows:

“The species in this category were so included because they met the relevant criteria developed (see Methods) but their inclusion in this category does not imply that international trade constitutes a major threat or that the threat applies throughout the species’ geographic range (Supplementary Discussion 5.1).”

interrogating these examples- the point I want to make is that the next steps for CITES can be informed by additional information in the IUCN Red List database. These steps should be further articulated in relation to Fig.4 and the authors should confirm whether any of these steps could be automated or, to apply the cautionary principle, further analysis of Red List data is best done manually as part of the CITES review process.

Regarding next steps, the Parties may wish to look at Red List assessments for particular species to look at what information is contained in the assessments. However, we expect (and encourage) Parties to consider consulting data and information sources beyond the IUCN Red List (e.g., recent literature) to determine if there is new, additional and/or supplementary information and data that they may wish to consider in decision-making. We now make this point more explicitly in the manuscript. Lines 266-269 now read as follows:

“Recognising the need for discussion with, and agreement from, the scientific committees (Supplementary Discussion 5.2), species could then be subsequently evaluated against the listing criteria, including drawing on additional information sources beyond the Red List, to determine which, if any, criteria they species meet (Fig. 4).”

Decision Letter, second revision:

14th March 2023

44Dear Dr. Challenger,

Thank you for submitting your revised manuscript "Identifying species likely threatened by international trade on the IUCN Red List can inform CITES trade measures" (NATECOLEVOL-220716931B). It has now been seen again by the original reviewers and their comments are below. The reviewers find that the paper has improved in revision, and therefore we'll be happy in principle to publish it in Nature Ecology & Evolution, pending minor revisions to satisfy the reviewers' final requests and to comply with our editorial and formatting guidelines.

[REDACTED]

Reviewer #1 (Remarks to the Author):

Dear Authors,

I have now reviewed the revised manuscript by Challenger et al as well as the comments from the reviewers and corresponding author replies.

I am comfortable with the authors response to my concerns and associated changes to their manuscript. Although I still have some concerns about the handling of national level trade, I believe the authors have sufficiently highlighted national level trade to ensure it isn't overlooked in the assessment process.

More over, no study is perfect. Considering this, one must ask if Challenger et al advances the field forward is a constructive and meaningful way. To this, my opinion is that they do. The most important aspect of their work is that they are creating a pipeline connecting IUCN Redlist to CITES; something that has been long asked for. I am confident that the authors will receive numerous responses to their work as well as follow-up studies that will either support or refute the proposed assessment framework. Such debate and/or consensus (if met) is a healthy and necessary step towards perfecting the framework they put forward.

On this point, dear authors, please ensure that all your data and associated code are available, organized, and interpretable to allow for efficient open-access and additional critique, improvement, and implementation in the future. Although I saw the supp data files available, since you are providing

45a workflow, having code available would be helpful for open use and oversight as well as adoption by others.

Reviewer #3 (Remarks to the Author):

The revised manuscript adequately addresses all previous comments.

Our ref: NATECOLEVOL-220716931B

22nd March 2023

Dear Dr. Challender,

Thank you for your patience as we've prepared the guidelines for final submission of your Nature Ecology & Evolution manuscript, "Identifying species likely threatened by international trade on the IUCN Red List can inform CITES trade measures" (NATECOLEVOL-220716931B). Please carefully follow the step-by-step instructions provided in the attached file, and add a response in each row of the table to indicate the changes that you have made. Please also check and comment on any additional marked-up edits we have proposed within the text. Ensuring that each point is addressed will help to ensure that your revised manuscript can be swiftly handed over to our production team.

****We would like to start working on your revised paper, with all of the requested files and forms, as soon as possible (preferably within two weeks). Please get in contact with us immediately if you anticipate it taking more than two weeks to submit these revised files.****

If you have not done so already, please alert us to any related manuscripts from your group that are under consideration or in press at other journals, or are being written up for submission to other journals (see: <https://www.nature.com/nature-research/editorial-policies/plagiarism#policy-on->

46duplicate-publication for details).

In recognition of the time and expertise our reviewers provide to Nature Ecology & Evolution's editorial process, we would like to formally acknowledge their contribution to the external peer review of your manuscript entitled "Identifying species likely threatened by international trade on the IUCN Red List can inform CITES trade measures". For those reviewers who give their assent, we will be publishing their names alongside the published article.

Nature Ecology & Evolution offers a Transparent Peer Review option for new original research manuscripts submitted after December 1st, 2019. As part of this initiative, we encourage our authors to support increased transparency into the peer review process by agreeing to have the reviewer comments, author rebuttal letters, and editorial decision letters published as a Supplementary item. When you submit your final files please clearly state in your cover letter whether or not you would like to participate in this initiative. Please note that failure to state your preference will result in delays in accepting your manuscript for publication.

Cover suggestions

As you prepare your final files we encourage you to consider whether you have any images or illustrations that may be appropriate for use on the cover of Nature Ecology & Evolution.

Nature Ecology & Evolution has now transitioned to a unified Rights Collection system which will allow our Author Services team to quickly and easily collect the rights and permissions required to publish your work. Approximately 10 days after your paper is formally accepted, you will receive an email in providing you with a link to complete the grant of rights. If your paper is eligible for Open Access, our Author Services team will also be in touch regarding any additional information that may be required to arrange payment for your article.

Please note that *Nature Ecology & Evolution* is a Transformative Journal (TJ). Authors may publish their research with us through the traditional subscription access route or make their paper immediately open access through payment of an article-processing charge (APC). Authors will not be

47required to make a final decision about access to their article until it has been accepted. [Find out more about Transformative Journals](https://www.springernature.com/gp/open-research/transformative-journals)

Authors may need to take specific actions to achieve [compliance with funder and institutional open access mandates](https://www.springernature.com/gp/open-research/funding/policy-compliance-faqs). If your research is supported by a funder that requires immediate open access (e.g. according to [Plan S principles](https://www.springernature.com/gp/open-research/plan-s-compliance)) then you should select the gold OA route, and we will direct you to the compliant route where possible. For authors selecting the subscription publication route, the journal's standard licensing terms will need to be accepted, including [self-archiving and license to publish](https://www.nature.com/nature-portfolio/editorial-policies/self-archiving-and-license-to-publish). Those licensing terms will supersede any other terms that the author or any third party may assert apply to any version of the manuscript.

[REDACTED]

[REDACTED]

Reviewer #1:
Remarks to the Author:
Dear Authors,

I have now reviewed the revised manuscript by Challenger et al as well as the comments from the reviewers and corresponding author replies.

I am comfortable with the authors response to my concerns and associated changes to their manuscript. Although I still have some concerns about the handling of national level trade, I believe the authors have sufficiently highlighted national level trade to ensure it isn't overlooked in the assessment process.

48More over, no study is perfect. Considering this, one must ask if Challender et al advances the field forward in a constructive and meaningful way. To this, my opinion is that they do. The most important aspect of their work is that they are creating a pipeline connecting IUCN Redlist to CITES; something that has been long asked for. I am confident that the authors will receive numerous responses to their work as well as follow-up studies that will either support or refute the proposed assessment framework. Such debate and/or consensus (if met) is a healthy and necessary step towards perfecting the framework they put forward.

On this point, dear authors, please ensure that all your data and associated code are available, organized, and interpretable to allow for efficient open-access and additional critique, improvement, and implementation in the future. Although I saw the supp data files available, since you are providing a workflow, having code available would be helpful for open use and oversight as well as adoption by others.

Reviewer #3:

Remarks to the Author:

The revised manuscript adequately addresses all previous comments.

Decision Letter, second revision:

Challender et al. – Identifying species likely threatened by international trade on the IUCN Red List can inform CITES trade measures

Reviewer comment (left hand column) and responses (right hand column).

Reviewer #1 (Remarks to the Author)	
I have now reviewed the revised manuscript by Challender et al as well as the comments from the reviewers and corresponding author replies.	Thank you for your further review and we are pleased that you consider that our manuscript has been improved.
I am comfortable with the authors response to my concerns and associated changes to their manuscript. Although I still have some concerns about the handling of national level	

49trade, I believe the authors have sufficiently highlighted national level trade to ensure it isn't overlooked in the assessment process.	
More over, no study is perfect. Considering this, one must ask if Challender et al advances the field forward is a constructive and meaningful way. To this, my opinion is that they do. The most important aspect of their work is that they are creating a pipeline connecting IUCN Redlist to CITES; something that has been long asked for. I am confident that the authors will receive numerous responses to their work as well as follow-up studies that will either support or refute the proposed assessment framework. Such debate and/or consensus (if met) is a healthy and necessary step towards perfecting the framework they put forward.	Thank you for recognising the contribution that this research makes.
On this point, dear authors, please ensure that all your data and associated code are available, organized, and interpretable to allow for efficient open-access and additional critique, improvement, and implementation in the future. Although I saw the supp data files available, since you are providing a workflow, having code available would be helpful for open use and oversight as well as adoption by others.	Thank you for this suggestion. We have provided data in two Supplementary Data files. The first includes all 21,745 species that were coded L/I/U, including their assigned category, as well as taxonomic information to species level and Red List Threat Category, and whether the taxa are included in CITES (I/II/III) or not. The second includes all species that have biological resource use (BRU) threat codes applied which our analyses

	indicate are not threatened by international trade but rather by use and/or trade at the sub-international level. This file also includes the current Appendix on which species are listed and otherwise indicates whether species are not currently included in CITES. We also provide the code for advanced automated coding. This has been uploaded to GitHub and the link included in Supplementary Methods Section 2.8.
Reviewer #2 (Remarks to the Author)	
1. The use of the IUCN red list as an early warning system, combined with the CITES trade appendices has been proposed before, and the authors do not demonstrate a clear pathway to policy - despite their recommendations.	Thank you for this feedback. We draw the reviewer’s attention to Lines 240-293 of our manuscript (including Fig. 4), where we detail how future iterations of our results could be produced rapidly and shared with the CITES Parties in various potential ways to inform decision making. Furthermore, we elaborate on this discussion in Supplementary Discussion 5.1 and 5.2. We believe that we have demonstrated that there is a clear pathway to policy. Moreover, at CITES CoP19 the Parties adopted Decision 19.186, which directs the CITES Standing Committee, in collaboration with the Animals and

	Plants Committee, to consider how to provide Parties requesting it with information from any relevant studies, analyses, or other sources on the identification of species at risk of extinction that are not yet regulated under CITES or may receive insufficient CITES regulation or that are or may be affected by international trade, working in coordination with the CITES Secretariat, CITES Parties, IUCN, UNEP-WCMC, FAO, regional competent authorities, and relevant experts as appropriate so that Parties may consider such information, as appropriate, in the preparation of listing proposals under Resolution Conf. 9.24 (Rev. CoP17) on the Criteria for amendment of Appendices I and II (see here). Following the adoption of this Decision, authors of this manuscript have met with the Chair of the CITES Animals Committee and are liaising with that Committee to explore how to feed our results in decision making in CITES. We have now included reference to this in the manuscript at Lines 240-243.
2. The circularity of their analyses (lines 202 – 205) is disingenuous, at times. The notion that CITES is not working because so many CITES listed species are threatened by trade does not consider the reasons/rationale for listing on	Thank you for this comment, which we have considered carefully. We do not suggest that CITES is not working and having considered the specific wording of this sentence and its placement in the

either database – or the correlation between information in both.	article – and the text that follows it – do not think readers would interpret it this way. The CITES strategic vision emphasises that international trade in species listed under the Convention should be legal and sustainable, such that it is not detrimental to them. The fact that 1,307 species remain threatened by international trade despite being included in the CITES Appendices – and therefore subject to the protection this affords them – does suggest that implementation could be stronger and the Convention more effective for these taxa (to the point that they are no longer threatened by international trade). But we do not suggest that these species are not benefitting at all; indeed, it is feasible that some are, but due to, for example, lag effects in species being reassessed on the Red List, it may not yet be readily apparent.
3. The use of the IUCN for systematic appraisals of sustainable trade also seems out-of-place.	The IUCN Red List of Threatened Species is the most authoritative source of information on the extinction risk to species globally. The Red List includes information on the use and trade of species. Respected as it is, we understand well the constraints, imperfections and inconsistencies in the Red List. In this study we have sought to systematically categorise species (starting with 38,245) as to whether they are likely threatened by international trade or not or if there is insufficient information to make this

	determination based on information in the Red List. We have been careful to include adequate detail on the methods used and have aimed to be fully transparent about the shortfalls in the Red List dataset in the caveats to our study. These are discussed in the methods section and in Supplementary Methods 2.1 and 2.3 and then in Supplementary Discussion 5.3.
Reviewer #3 (Remarks to the Author)	
The revised manuscript adequately addresses all previous comments.	Thank you for confirming that this is the case.

Final Decision Letter:

7th June 2023

Dear Dan,

Thanks for your patience with all the final stages before acceptance of your paper. I'm pleased to now be able to send you this email to inform you that your Article entitled "Identifying species likely threatened by international trade on the IUCN Red List can inform CITES trade measures", has now been accepted for publication in Nature Ecology & Evolution.

Over the next few weeks, your paper will be copyedited to ensure that it conforms to Nature Ecology and Evolution style. Once your paper is typeset, you will receive an email with a link to choose the appropriate publishing options for your paper and our Author Services team will be in touch regarding any additional information that may be required

You will not receive your proofs until the publishing agreement has been received through our system

54Due to the importance of these deadlines, we ask you please us know now whether you will be difficult to contact over the next month. If this is the case, we ask you provide us with the contact information (email, phone and fax) of someone who will be able to check the proofs on your behalf, and who will be available to address any last-minute problems . Once your paper has been scheduled for online publication, the Nature press office will be in touch to confirm the details.

Acceptance of your manuscript is conditional on all authors' agreement with our publication policies (see www.nature.com/authors/policies/index.html). In particular your manuscript must not be published elsewhere and there must be no announcement of the work to any media outlet until the publication date (the day on which it is uploaded onto our web site).

Please note that *Nature Ecology & Evolution* is a Transformative Journal (TJ). Authors may publish their research with us through the traditional subscription access route or make their paper immediately open access through payment of an article-processing charge (APC). Authors will not be required to make a final decision about access to their article until it has been accepted. [Find out more about Transformative Journals](https://www.springernature.com/gp/open-research/transformative-journals)

Authors may need to take specific actions to achieve [compliance](https://www.springernature.com/gp/open-research/funding/policy-compliance-faqs) with funder and institutional open access mandates. If your research is supported by a funder that requires immediate open access (e.g. according to [Plan S principles](https://www.springernature.com/gp/open-research/plan-s-compliance)) then you should select the gold OA route, and we will direct you to the compliant route where possible. For authors selecting the subscription publication route, the journal's standard licensing terms will need to be accepted, including [self-archiving-and-license-to-publish](https://www.nature.com/nature-portfolio/editorial-policies/self-archiving-and-license-to-publish). Those licensing terms will supersede any other terms that the author or any third party may assert apply to any version of the manuscript.

We welcome the submission of potential cover material (including a short caption of around 40 words) related to your manuscript; suggestions should be sent to Nature Ecology & Evolution as electronic files (the image should be 300 dpi at 210 x 297 mm in either TIFF or JPEG format). Please note that such pictures should be selected more for their aesthetic appeal than for their scientific content, and that colour images work better than black and white or grayscale images. Please do not try to design a cover with the Nature Ecology & Evolution logo etc., and please do not submit composites of images related to your work. I am sure you will understand that we cannot make any promise as to whether any of your suggestions might be selected for the cover of the journal.

You can generate the link yourself when you receive your article DOI by entering it here: <http://authors.springernature.com/share>.

Thank you for choosing NEE to publish this work, and for all your patience along the way. I'm very much looking forward to seeing it published soon.

[REDACTED]

P.S. Click on the following link if you would like to recommend Nature Ecology & Evolution to your librarian <http://www.nature.com/subscriptions/recommend.html#forms>

** Visit the Springer Nature Editorial and Publishing website at http://editorial-jobs.springernature.com?utm_source=ejp_NEcoE_email&utm_medium=ejp_NEcoE_email&utm_campaign=ejp_NEcoE for more information about our career opportunities. If you have any questions please click [here](mailto:editorial.publishing.jobs@springernature.com).**